# No-Regret Online Prediction with Strategic Experts

**Omid Sadeghi**     **Maryam Fazel**
University of Washington
Seattle, WA 98195
{omids,mfazel@uw.edu}

## Abstract

We study a generalization of the online binary prediction with expert advice framework where at each round, the learner is allowed to pick $m \geq 1$ experts from a pool of $K$ experts and the overall utility is a modular or submodular function of the chosen experts. We focus on the setting in which experts act strategically and aim to maximize their influence on the algorithm's predictions by potentially misreporting their beliefs about the events. Among others, this setting finds applications in forecasting competitions where the learner seeks not only to make predictions by aggregating different forecasters but also to rank them according to their relative performance. Our goal is to design algorithms that satisfy the following two requirements: 1) *Incentive-compatible*: Incentivize the experts to report their beliefs truthfully, and 2) *No-regret*: Achieve sublinear regret with respect to the true beliefs of the best-fixed set of $m$ experts in hindsight. Prior works have studied this framework when $m = 1$ and provided incentive-compatible no-regret algorithms for the problem. We first show that a simple reduction of our problem to the $m = 1$ setting is neither efficient nor effective. Then, we provide algorithms that utilize the specific structure of the utility functions to achieve the two desired goals.

## 1   Introduction

Learning from a constant flow of information is one of the most prominent challenges in machine learning. In particular, online learning requires the learner to iteratively make decisions and at the time of making each decision, the outcome associated with it is unknown to the learner. The *experts problem* is perhaps the most well-known problem in online learning [1, 2, 3, 4]. In this problem, the learner aims to make predictions about a sequence of $T$ binary events. To do so, the learner has access to the advice of $K$ experts who each have internal beliefs about the likelihood of each event. At each round $t \in [T]$, the learner has to choose one among the advice of $K$ experts and upon making her choice, the $t$-th binary event is realized and a loss bounded between zero and one is revealed. The goal of the learner is to have no regret, i.e., to perform as well as the best-fixed expert in hindsight. In many applications, however, the experts are strategic and wish to be selected by the learner as often as possible. To this end, they may strategically misreport their beliefs about the events. For instance, FiveThirtyEight[1] aggregates different pollsters according to their past performance to make a single prediction for elections and sports matches. To do so, FiveThirtyEight maintains publicly available pollster ratings[2]. A low rating can be harmful to the pollster's credibility and adversely impact their revenue opportunities in the future. Therefore, instead of maximizing their expected performance by reporting their predictions truthfully, the pollsters may decide to take risks and report more extreme beliefs to climb higher on the leaderboard. Therefore, it is important to design algorithms that not only achieve *no-regret* but also motivate the experts to report their true beliefs (*incentive-compatible*).

---

[1]https://fivethirtyeight.com/

[2]https://projects.fivethirtyeight.com/pollster-ratings/

37th Conference on Neural Information Processing Systems (NeurIPS 2023).

Otherwise, the quality of the learner's predictions may be harmed.

As we have mentioned in Section 1.1, all the previous works on this topic have focused on the standard experts problem where the goal is to choose a *single* expert among the $K$ experts. In the offline setting, this is equivalent to a forecasting competition in which only the single highest-ranked forecaster wins and receives prizes. However, in many applications, a *set* of top-performing forecasters are awarded perks and benefits. For instance, in the Good Judgement Project, a recent geopolitical forecasting tournament, the top $2\%$ of forecasters were given the "superforecaster" status and received benefits such as paid conference travel and employment opportunities [5]. Similarly, in the latest edition of Kaggle's annual machine learning competition to predict the match outcomes of the NCAA March Madness college basketball tournament (called "March Machine Learning Mania 2023"[3]), the top 8 forecasters on the leaderboard received monetary prizes.

Variants of the $m$-experts problem have been previously studied in [6, 7, 8, 9], however, all of these works focused only on providing no-regret algorithms for the problem and the incentive compatibility considerations were not taken into account. To the best of our knowledge, this is the first work that focuses on the strategic $m$-experts problem where the experts may misreport their true beliefs to increase their chances of being chosen by the learner.

For the setting with modular utilities, perhaps the simplest approach to learning well compared to the best-fixed set of $m$ experts while maintaining incentive compatibility is to run the incentive-compatible WSU algorithm of [10] for the standard experts problem (the setting with $m = 1$) over the set of $\binom{K}{m}$ meta-experts where each meta-expert corresponds to one of the sets of size $m$. This approach has two major drawbacks: 1) There are exponentially many meta-experts and maintaining weights per each meta-expert and running the WSU algorithm is computationally expensive, and 2) The dependence of the regret bound on $m$ is sub-optimal. Therefore, for our setting, it is preferable to design algorithms that are tailored for the $m$-experts problem.

## 1.1 Related work

Prior works have studied the experts problem under incentive compatibility considerations for two feedback models: In the full information setting, the learner observes the reported prediction of all experts at each round. In the partial information setting, however, the learner is restricted to choosing a single expert at each round and does not observe the prediction of other experts. [11] considered algorithms that maintain weights over the experts and choose experts according to these weights. They assumed that experts' incentives are only affected by the unnormalized weights of the algorithm over the experts. However, since the probability of an expert being chosen by the learner equals her normalized weight, the aforementioned assumption might not be suitable. Later on, [10] made the assumption that at each round $t \in [T]$, incentives are tied to the expert's normalized weight (i.e., the probability of being chosen at round $t+1$) and studied this problem under both feedback models. They proposed the WSU and WSU-UX algorithms for the full information and partial information settings respectively where both algorithms are incentive-compatible and they obtained $\mathcal{O}(\sqrt{T\ln K})$ and $\mathcal{O}(T^{2/3}(K\ln K)^{1/3})$ regret bounds for the algorithms. Then, [12] considered *non-myopic* strategic experts where the goal of each expert is to maximize a conic combination of the probabilities of being chosen in all subsequent rounds (not just the very next round). They showed that the well-known Follow the Regularized Leader (FTRL) algorithm with the negative entropy regularizer obtains a regret bound of $\mathcal{O}(\sqrt{T \ln K})$ while being $\Theta(\frac{1}{\sqrt{T}})$ approximately incentive-compatible, i.e., it is a strictly dominated strategy for any expert to make reports $\Theta(\frac{1}{\sqrt{T}})$ distant from their true beliefs.

For the non-strategic $m$-experts problem with modular utilities, [7] proposed the Component Hedge (CH) algorithm and obtained a regret bound of $\sqrt{2m\ell^* \ln(\frac{K}{m})} + m \ln(\frac{K}{m})$ where $\ell^*$ is the cumulative loss of the best-chosen set in hindsight. They also gave a matching lower bound for this problem. [8] studied the FTPL algorithm with Gaussian noise distribution and provided an $\mathcal{O}(m\sqrt{T \ln(\frac{K}{m})})$ regret bound for this setting. For the non-strategic $m$-experts problem with submodular utility functions, [13] proposed the online distorted greedy algorithm (with dual averaging algorithm as the algorithm $\mathcal{A}_i$ for $i = 1, \ldots, m$) whose regret bound is $\mathcal{O}(\sqrt{mT \ln(\frac{K}{m})})$. More recently, [9] studied the $m$-experts problem under various choices of the utility function (sum-reward, max-reward, pairwise-reward and monotone reward). In particular, for the setting with modular utilities (sum-reward), they proposed

---

[3]https://www.kaggle.com/competitions/march-machine-learning-mania-2023/

an algorithm that matches the optimal regret bound of the CH algorithm while being computationally more efficient.

## 1.2 Contributions

In this paper, we focus on a generalization of the experts problem (called "$m$-experts problem") where at each round, instead of picking a single expert, we are allowed to pick $m \geq 1$ experts and our utility is either a modular or submodular function of the chosen experts. In particular, at round $t \in [T]$ and for a set of experts $S_t \subseteq [K]$, the utility function is defined as $f_t(S_t) = \frac{|S_t|}{m} - \frac{1}{m} \sum_{i \in S_t} \ell_{i,t}$ and $f_t(S_t) = 1 - \prod_{i \in S_t} \ell_{i,t}$ in the modular and submodular cases respectively where $\ell_{i,t} \in [0,1]$ is the loss of expert $i$ at round $t$. The goal is to design algorithms that perform as well as the best-fixed set of $m$ experts in hindsight (*no-regret*) and incentivize the experts to report their beliefs about the events truthfully (*incentive-compatible*).

Towards this goal, we build upon the study of the Follow the Perturbed Leader (FTPL) algorithm for the $m$-experts problem with modular utility functions by [8] and derive a sufficient condition for the perturbation distribution to guarantee approximate incentive compatibility. Furthermore, we show how this condition is related to the commonly used bounded hazard rate assumption for noise distribution. In particular, we show that while FTPL with Gaussian perturbations is not incentive-compatible, choosing Laplace or hyperbolic noise distribution guarantees approximate incentive compatibility.

Moreover, inspired by Algorithm 1 of [13] for online monotone submodular maximization subject to a matroid constraint, we first introduce a simpler algorithm (called the "online distorted greedy algorithm") for the special case of cardinality constraints. This algorithm utilizes $m$ incentive-compatible algorithms for the standard experts problem (i.e., $m = 1$ setting) and outputs their combined predictions. We provide $(1 - \frac{c}{e})$-regret bounds for the algorithm where $c \in [0,1]$ is the average curvature of the submodular utility functions. Therefore, applying the algorithm to the setting where the utility functions are modular (i.e., $c = 0$), the approximation ratio is 1. For submodular utility functions, the algorithm achieves the optimal $1 - \frac{c}{e}$ approximation ratio.

Finally, we validate our theoretical results through experiments on data gathered from a forecasting competition run by FiveThirtyEight in which forecasters make predictions about the match outcomes of the recent 2022–2023 National Football League (NFL).

## 2 Preliminaries

**Notation.** The set $\{1, 2, \ldots, n\}$ is denoted by $[n]$. For vectors $x_t, y \in \mathbb{R}^n$, $x_{i,t}$ and $y_i$ denote their $i$-th entry respectively. Similarly, for a matrix $A \in \mathbb{R}^{n \times n}$, we use $A_{i,j}$ to indicate the entry in the matrix's $i$-th row and $j$-th column. The inner product of two vectors $x, y \in \mathbb{R}^n$ is denoted by either $\langle x, y \rangle$ or $x^T y$. For a set function $f$, we use $f(j|A)$ to denote $f(A \cup \{j\}) - f(A)$.

A set function $f$ defined over the ground set $V$ is monotone if for all $A \subseteq B \subseteq V$, $f(A) \leq f(B)$ holds. $f$ is called submodular if for all $A \subseteq B \subset V$ and $j \notin B$, $f(j|A) \geq f(j|B)$ holds. In other words, the marginal gain of adding the item $j$ decreases as the base set gets larger. This property is known as the diminishing returns property. If equality holds in the above inequality for all $A$, $B$, and $j$, the function is called modular. As mentioned in Section 1.2, at round $t \in [T]$ and for a set of experts $S \subseteq [K]$, the submodular utility function is defined as $f_t(S_t) = 1 - \prod_{i \in S_t} \ell_{i,t}$ where $\ell_{i,t} \in [0,1]$ is the loss of expert $i$ at round $t$. To show that this function is submodular, note that we have:

$$f(A \cup \{j\}) - f(A) = (1 - \ell_{j,t}) \prod_{i \in A} \ell_{i,t} \geq (1 - \ell_{j,t}) \prod_{i \in B} \ell_{i,t} = f(B \cup \{j\}) - f(B),$$

where the inequality follows from $\ell_{i,t} \in [0,1]$ for $i \in B \setminus A$.

For a normalized monotone submodular set function $f$ (i.e., $f(\emptyset) = 0$), the curvature $c_f$ is defined as [14]:

$$c_f = 1 - \min_{j \in V} \frac{f(j|V \setminus \{j\})}{f(\{j\})}.$$

It is easy to see that $c_f \in [0,1]$ always holds. $c_f \leq 1$ is due to monotonicity of $f$ and $c_f \geq 0$ follows from $f$ being submodular. Curvature characterizes how submodular the function is. If $c_f = 0$, the function is modular, and larger values of $c_f$ correspond to the function exhibiting a stronger diminishing returns structure.

# 3 $m$-experts problem

We introduce the $m$-experts problem in this section. In this problem, there are $K$ experts available and each expert makes probabilistic predictions about a sequence of $T$ binary outcomes. At round $t \in [T]$, expert $i \in [K]$ has a private belief $b_{i,t} \in [0, 1]$ about the outcome $r_t \in \{0, 1\}$, where $r_t$ and $\{b_{i,t}\}_{i=1}^K$ are chosen arbitrarily and potentially adversarially. Expert $i$ reports $p_{i,t} \in [0, 1]$ as her prediction to the learner. Then, the learner chooses a set $S_t$ containing $m$ of the experts. Upon committing to this action, the outcome $r_t$ is revealed, and expert $i$ incurs a loss of $\ell_{i,t} = \ell(b_{i,t}, r_t)$ where $\ell : [0, 1] \times \{0, 1\} \to [0, 1]$ is a bounded loss function. In this paper, we focus on the quadratic loss function defined as $\ell(b, r) = (b - r)^2$. The utility of the learner at round $t$ is one of the following:

- **Modular utility function:** $f_t(S_t) = \frac{|S_t|}{m} - \frac{1}{m} \sum_{i \in S_t} \ell_{i,t} = \frac{|S_t|}{m} - \frac{1}{m} \sum_{i \in S_t} \ell(b_{i,t}, r_t)$.
- **Submodular utility function:** $f_t(S_t) = 1 - \prod_{i \in S_t} \ell_{i,t} = 1 - \prod_{i \in S_t} \ell(b_{i,t}, r_t)$.

It is easy to see that $f_t$ is monotone in both cases and $f_t(S_t) \in [0, 1]$ holds. Note that the utility at each round is defined with respect to the true beliefs of the chosen experts rather than their reported beliefs.

The goal of the learner is twofold:

1) Minimize the $\alpha$-regret defined as $\alpha\text{-}R_T = \mathbb{E}\big[\alpha \max_{S \subseteq [K]:|S|=m} \sum_{t=1}^T f_t(S) - \sum_{t=1}^T f_t(S_t)\big]$, where the expectation is taken with respect to the potential randomness of the algorithm. For the modular utility function, $\alpha = 1$ and for the submodular setting, we set $\alpha = 1 - \frac{c_f}{e}$ (where $f = \sum_{t=1}^T f_t$) which is the optimal approximation ratio for any algorithm making polynomially many queries to the objective function.

2) Incentivize experts to report their private beliefs truthfully. To be precise, at each round $t \in [T]$, each expert $i \in [K]$ acts strategically to maximize their probability of being chosen at round $t + 1$ and the learner's algorithm is called incentive-compatible if expert $i$ maximizes this probability by reporting $p_{i,t} = b_{i,t}$. To be precise, we define the incentive compatibility property below.

**Definition 1.** *An online learning algorithm is incentive-compatible if for every $t \in [T]$, every expert $i \in [K]$ with belief $b_{i,t}$, every report $p_{i,t}$, reports of other experts $p_{-i,t}$, every history of reports $\{p_{j,s}\}_{j \in [K], s < t}$, and outcomes $\{r_s\}_{s < t}$, we have:*

$$\mathbb{E}_{r_t \sim Bern(b_{i,t})}\big[\pi_{i,t+1} \mid b_{i,t}, p_{-i,t}, \{p_{j,s}\}_{j \in [K], s < t}, \{r_s\}_{s < t}\big]$$
$$\geq \mathbb{E}_{r_t \sim Bern(b_{i,t})}\big[\pi_{i,t+1} \mid p_{i,t}, p_{-i,t}, \{p_{j,s}\}_{j \in [K], s < t}, \{r_s\}_{s < t}\big],$$

*where $Bern(b)$ denotes a Bernoulli distribution with probability of success $b$ and $\pi_{i,t+1}$ is the probability of expert $i$ being chosen at round $t + 1$.*

In other words, an online learning algorithm is incentive-compatible if it is in the best interest of the experts to report their private beliefs truthfully to maximize the probability of being chosen for the next round.

As mentioned earlier, we focus on the quadratic loss function in this paper. The quadratic loss function is an instance of proper loss functions [15], i.e., the following holds:

$$\mathbb{E}_{r \sim \text{Bern}(b)}[\ell(p, r)] \geq \mathbb{E}_{r \sim \text{Bern}(b)}[\ell(b, r)] \ \forall p \neq b,$$

i.e., each expert minimizes her expected loss (according to their true belief $b$) by reporting truthfully.

## 3.1 Motivating applications

Several interesting motivating applications could be cast into our framework. We mention two classes of such applications below.

- *Forecasting competitions*: In this problem, there are a set of $K$ forecasters who aim to predict the outcome of sports games or elections (between two candidates). At each round $t \in [T]$, information on the past performance of the two opposing teams or candidates is revealed and forecasters provide a probabilistic prediction (as a value between $[0, 1]$) about which team or candidate will win. The learner can choose up to $m$ forecasters at each round and her utility is simply the average of the utilities of chosen experts.

- *Online paging problem with advice* [16]: There is a library $\{1, \ldots, N\}$ of $N$ distinct files. A cache with limited storage capacity can store at most $m$ files at any time. At each round $t \in [T]$, a user arrives and requests one file. The learner has access to a pool of $K$ experts where each expert $i \in [K]$ observes the user history and makes a probabilistic prediction $p_{i,t} \in [0, 1]^N$ for the next file

request (where $1^T p_{i,t} = 1$). For instance, $p_{i,t} = e_j$ if expert $i$ predicts the file $j \in [N]$ where $e_j$ is the $j$-th standard basis vector. Also, $r_t = e_j$ if the $j$-th file is requested at round $t \in [T]$. The learner can choose $m$ of these experts at each round and put their predictions in the cache. The learner's prediction for round $t$ is correct if and only if one of the $m$ chosen experts has correctly predicted the file. Thus, the loss of expert $i$ can be formulated as $\ell_{i,t} = \|p_{i,t} - r_t\|_2^2$ and the utility at round $t$ could be written as $f_t(S_t) = 2 - \prod_{i \in S_t} \ell_{i,t}$ which is exactly our submodular utility function. Note that this is a slight generalization of our framework where instead of binary outcomes, we consider nonbinary (categorical) outcomes. All our results could be easily extended to this setting as well.

### 3.2 Naive approach

The WSU algorithm of [10] for the standard experts problem is derived by drawing a connection between online learning and wagering mechanisms. The framework of one-shot wagering mechanisms was introduced by [17] and is as follows: There are $K$ experts and each expert $i \in [K]$ holds a belief $b_i \in [0, 1]$ about the likelihood of an event. Expert $i$ reports a probability $p_i \in [0, 1]$ and a wager $\omega_i \geq 0$. A wagering mechanism $\Gamma$ is a mapping from the reports $p = (p_1, \ldots, p_K)$, wagers $\omega = (\omega_1, \ldots, \omega_K)$ and the realization $r$ of the binary event to the payments $\Gamma_i(p, \omega, r)$ to expert $i$. It is assumed that $\Gamma_i(p, \omega, r) \geq 0 \; \forall i \in [K]$, i.e., no expert loses more than her wager. A wagering mechanism is called budget-balanced if $\sum_{i=1}^K \Gamma_i(p, \omega, r) = \sum_{i=1}^K \omega_i$. [17, 18] introduced a class of incentive-compatible budget-balanced wagering mechanisms called the Weighted Score Wagering Mechanisms (WSWMs) which is defined as follows: For a fixed proper loss function $\ell$ bounded in $[0, 1]$, the payment to expert $i$ is

$$\Gamma_i(p, \omega, r) = \omega_i \Big(1 - \ell(p_i, r) + \sum_{j=1}^K \omega_j \ell(p_j, r)\Big).$$

The proposed algorithm in [10] is called Weighted-Score Update (WSU) and the update rule for the weights of the experts $\{\pi_t\}_{t=1}^T$ is the following:

$$\pi_{i,t+1} = \eta \Gamma_i(p_t, \pi_t, r_t) + (1 - \eta)\pi_{i,t},$$

where $\pi_{i,1} = \frac{1}{K} \; \forall i \in [K]$. In other words, the normalized weights of the experts at round $t$ are interpreted as the wager of the corresponding expert, and the normalized weights at round $t + 1$ are derived using a convex combination of the weights at the previous round and the payments in WSWM. Note that since WSWM is budget-balanced, the derived weights at each round automatically sum to one and there is no need to normalize the weights (which might break the incentive compatibility). Also, considering the incentive compatibility of WSWM, the WSU algorithm is incentive-compatible as well.

The update rule of WSU could be rewritten as follows:

$$\pi_{i,t+1} = \eta \pi_{i,t}\Big(1 - \ell_{i,t} + \sum_{j=1}^K \pi_{j,t}\ell_{j,t}\Big) + (1 - \eta)\pi_{i,t} = \pi_{i,t}(1 - \eta L_{i,t}),$$

where $L_{i,t} = \ell_{i,t} - \sum_{j=1}^K \pi_{j,t}\ell_{j,t}$. Therefore, the WSU update rule is similar to that of the Multiplicative Weights Update (MWU) algorithm [19] with the relative loss $L_{i,t}$ instead of $\ell_{i,t}$ in the formula.

A simple approach to solving the $m$-experts problem with modular utilities ($\ell_{S,t} = \frac{1}{m} \sum_{j \in S} \ell_{j,t}$) is to define an "expert" for each of the possible $\binom{K}{m}$ sets of size $m$ and apply the incentive-compatible WSU algorithm of [10] for the standard experts problem to this setting. Note that we still define incentive compatibility with respect to individual experts (instead of the $\binom{K}{m}$ meta-experts). To be precise, we define $\pi_{i,t} = \sum_{S:|S|=m,i \in S} \pi_{S,t}$. We can show the following:

$$\pi_{i,t+1} = \sum_{S:|S|=m,i \in S} \pi_{S,t+1} = \pi_{i,t}(1 - \frac{\eta}{m}L_{i,t}) - \frac{\eta}{m}\sum_{s \neq i}\Big(\sum_{S:|S|=m,\{i,s\}\subseteq S} \pi_{S,t}\Big)\ell_{s,t}.$$

Given that $\pi_{i,t+1}$ is linear in $L_{i,t}$, $L_{i,t} = \ell_{i,t} - \sum_{j=1}^K \pi_{j,t}\ell_{j,t}$ is linear in $\ell_{i,t}$, and the loss function is proper, we can conclude that incentive compatibility holds in this setting as well.

We summarize the result of this approach in the theorem below.

**Theorem 1.** *If we apply the WSU algorithm of [10] to a standard experts problem with $\binom{K}{m}$ experts corresponding to each $S$ with $|S| = m$, and set $\eta = \sqrt{\frac{m \ln(\frac{Ke}{m})}{T}}$, the algorithm is incentive-compatible and its regret is bounded as follows:*

$$\mathbb{E}[1\text{-}R_T] \leq \mathcal{O}(\sqrt{mT \ln(\frac{K}{m})}).$$

This approach has two major drawbacks:
1) Computational complexity of maintaining weights for each $\binom{K}{m}$ feasible sets. In particular, we have to do exponentially many queries to the objective function at each round to update these weights.
2) The regret bound has a $\sqrt{m}$ dependence on the number of experts $m$ that is suboptimal.
In the subsequent sections, we propose two efficient algorithmic frameworks that exploit the modular or submodular structure of the utility function and obtain the desired regret and incentive compatibility guarantees.

## 4   Follow the Perturbed Leader (FTPL) algorithm

In this section, we study the well-known Follow the Perturbed Leader (FTPL) algorithm for the $m$-experts problem with modular utility functions and study its regret and incentive compatibility guarantees. The algorithm is as follows: At each round $t \in [T]$, we first take $K$ i.i.d. samples $\{\gamma_{i,t}\}_{i=1}^K$ from the noise distribution $\mathcal{D}$. In particular, we focus on zero-mean symmetric noise distributions from the exponential family, i.e., $f(\gamma_{i,t}) \propto \exp(-\nu(\gamma_{i,t}))$ where $\nu : \mathbb{R} \to \mathbb{R}_+$ is symmetric about the origin. At round $t$, we simply keep track of $\sum_{s=1}^{t-1} \ell_{i,s} + \eta\gamma_{i,t}$ for each $i$ (where $\eta$ is the step size) and pick the $m$ experts for whom this quantity is the smallest. [8] previously studied the FTPL algorithm for a class of problems that includes the $m$-experts problem. However, they only focused on the setting with zero-mean Gaussian perturbations. In contrast, we not only extend this analysis to all zero-mean symmetric noise distributions from the exponential family, but we also analyze the incentive compatibility guarantees of the algorithm and determine a sufficient condition for the perturbation distribution under which the algorithm is approximately incentive-compatible. This condition is provided below.

**Condition 1.** *For all $z \in \mathbb{R}$, $|\nu'(z)| \leq B$ holds for some constant $B > 0$.*

We have $\nu(z) = |z|$ for Laplace distribution. Therefore, $\nu'(z) = \text{sign}(z)$ and $B = 1$. For symmetric hyperbolic distribution, $\nu(z) = \sqrt{1 + z^2}$ holds. So, $|\nu'(z)| = \frac{|z|}{\sqrt{1+z^2}} \leq 1$ and $B = 1$.
Condition 1 is closely related to a boundedness assumption on the hazard rate of the perturbation distribution. We first define the hazard rate below.

**Definition 2.** *The hazard rate of $\mathcal{D}$ at $z \in \mathbb{R}$ is defined as*

$$haz_{\mathcal{D}}(z) = \frac{f_{\mathcal{D}}(z)}{1 - F_{\mathcal{D}}(z)},$$

*where $f_{\mathcal{D}}$ and $F_{\mathcal{D}}$ are the probability density function (pdf) and the cumulative density function (cdf) of the noise distribution $\mathcal{D}$. The maximum hazard rate of $\mathcal{D}$ is $haz_{\mathcal{D}} = \sup_{z \in \mathbb{R}} haz_{\mathcal{D}}(z)$.*

The hazard rate is a statistical tool used in survival analysis that measures how fast the tail of a distribution decays. The theorem below shows the connection between Condition 1 and the bounded hazard rate assumption.

**Theorem 2.** *If Condition 1 holds for the perturbation distribution $\mathcal{D}$ with the constant $B > 0$, we have $haz_{\mathcal{D}} \leq B$.*

However, there are distributions with bounded hazard rates for which $\max_z |\nu'(z)|$ is unbounded (i.e., Condition 1 does not hold). For instance, consider the standard Gumbel distribution. In this case, $\nu(z) = z + \exp(-z)$. Therefore, we have $\nu'(z) = 1 - \exp(-z)$. So, if $z \to -\infty$, $|\nu'(z)| \to \infty$. Therefore, Condition 1 is strictly stronger than the bounded hazard rate assumption for the noise distribution $\mathcal{D}$.
We show how Condition 1 guarantees an approximate notion of incentive compatibility for FTPL.

**Theorem 3.** *For the FTPL algorithm with a noise distribution satisfying Condition 1 with a constant $B > 0$, at round $t \in [T]$, for an expert $i \in [K]$, the optimal report from the expert's perspective $p_{i,t}^*$ is at most $\frac{2B}{\eta - 2B}$ away from her belief $b_{i,t}$, i.e., the following holds:*

$$|p_{i,t}^* - b_{i,t}| \leq \frac{2B}{\eta - 2B}.$$

Note that while we focused on the incentive structure in which at each round $t \in [T]$, experts wish to maximize their probability of being chosen at round $t + 1$, the same argument could be applied to a more general setting where the goal is to maximize a conic combination of probabilities of being chosen at all subsequent round $s > t$. Therefore, FTPL is approximately incentive compatible with respect to this more general incentive structure as well.

Theorem 3 allows us to bound the regret of the FTPL algorithm with respect to the true beliefs of the experts. First, note that FTPL obtains the following bound with respect to the reported beliefs of the experts.

**Theorem 4.** *For the FTPL algorithm with noise distribution $\mathcal{D}$ satisfying Condition 1 with the constant $B > 0$, if we set $\eta = \sqrt{\frac{BT}{\ln(\frac{K}{m})}}$, the following holds:*

$$\mathbb{E}[\frac{1}{m} \sum_{t=1}^{T} \sum_{i \in S_t} \ell(p_{i,t}, r_t) - \min_{S:|S|=m} \frac{1}{m} \sum_{t=1}^{T} \sum_{j \in S} \ell(p_{j,t}, r_t)] \leq \mathcal{O}(\sqrt{BT \ln(\frac{K}{m})}).$$

Using the result of Theorem 3, we have $|p_{i,t} - b_{i,t}| = |p_{i,t}^* - b_{i,t}| \leq \frac{2B}{\eta - 2B}$. Moreover, one can easily show that the quadratic loss function is 2-Lipschitz. Therefore, for all $t \in [T]$ and $i \in [K]$, we have:

$$|\ell(p_{i,t}, r_t) - \ell(b_{i,t}, r_t)| \leq \frac{4B}{\eta - 2B}.$$

Putting the above results together, we can obtain the following regret bound for the FTPL algorithm.

$$\mathbb{E}[1\text{-}R_T] = \mathbb{E}[\frac{1}{m} \sum_{t=1}^{T} \sum_{i \in S_t} \ell(b_{i,t}, r_t) - \min_{S:|S|=m} \frac{1}{m} \sum_{t=1}^{T} \sum_{j \in S} \ell(b_{j,t}, r_t)] \leq \mathcal{O}(\sqrt{BT \ln(\frac{K}{m})}) + \frac{8BT}{\eta - 2B}.$$

Given that $\eta = \sqrt{\frac{BT}{\ln(\frac{K}{m})}}$ in Theorem 4, the expected regret bound is $\mathcal{O}(\sqrt{BT \ln(\frac{K}{m})})$. This result is summarized in the following theorem.

**Theorem 5.** *For the FTPL algorithm with noise distribution $\mathcal{D}$ satisfying Condition 1 with the constant $B > 0$, if we set $\eta = \sqrt{\frac{BT}{\ln(\frac{K}{m})}}$, the regret bound is $\mathcal{O}(\sqrt{BT \ln(\frac{K}{m})})$.*

In order to ensure approximate incentive compatibility, the probability density function of the noise distribution $f$ needs to be such that $\frac{f(z)}{f(z+1)}$ does not grow to infinity for very large $z$. One way to enforce this condition is via a Lipschitzness assumption on $\ln f$. Condition 1 implies that $\ln f$ is $B$-Lipschitz. That is why smaller values of $B$ lead to better approximate incentive compatibility which in turn results in smaller regret bounds (given that the term $4TC$ appears in the regret bound where $C$ is the bound on the approximate incentive-compatibility derived in Theorem 3).

We can use the FTPL algorithm to obtain results for the partial information setting as well. [20] showed that if the hazard rate of the noise distribution is bounded by $B$, applying the FTPL algorithm to the partial information setting for the 1-expert problem leads to $\mathcal{O}(\sqrt{BKT \ln K})$ regret bounds. Using the result of Theorem 2, we know that if Condition 1 holds, the hazard rate is bounded. Therefore, if the noise distribution satisfies Condition 1, FTPL applied to the $m$-experts problem is approximately incentive-compatible and achieves $\mathcal{O}(\sqrt{BKT \ln(\frac{K}{m})})$ regret bound.

## 5   Online distorted greedy algorithm

In this section, we study the setting where the utility function is submodular. In this case, we have $f_t(S_t) = 1 - \prod_{i \in S_t} \ell_{i,t} = 1 - \prod_{i \in S_t} \ell(b_{i,t}, r_t)$. The problem in this setting could be

written as an online monotone submodular maximization problem subject to a cardinality constraint of size $m$. [21] proposed the online greedy algorithm for this problem whose $(1 - \frac{1}{e})$-regret is bounded by $\mathcal{O}(\sqrt{mT \ln K})$. The algorithm works as follows: There are $m$ instantiations $\mathcal{A}_1, \ldots, \mathcal{A}_m$ of no-regret algorithms for the 1-expert problem. At each round $t \in [T]$, $\mathcal{A}_i$ selects an expert $v_{i,t} \in [K]$ and the set $S_t = \{v_{1,t}, \ldots, v_{m,t}\}$ is selected. $\mathcal{A}_i$ observes the reward $f_t(v_{i,t}|S_{i-1,t})$ where $S_{j,t} = \{v_{1,t}, \ldots, v_{j,t}\}$.

Inspired by Algorithm 1 of [13] for online monotone submodular maximization subject to a matroid constraint, we propose the online distorted greedy in Algorithm 1 for the special case of a cardinality constraint. The algorithm is similar to the online greedy algorithm of [21] discussed above. However, in the online distorted greedy algorithm, after choosing the set $S_t$ and observing the function $f_t$, we first compute the modular lower bound $h_t$ defined as $h_t(S) = \sum_{i \in S} f_t(i|[K] \setminus \{i\})$. We define $g_t = f_t - h_t$. Note that $g_t$ is monotone submodular as well. The reward of $\mathcal{A}_i$ for choosing $v_{i,t}$ at round $t$ is $(1 - \frac{1}{m})^{m-i} g_t(v_{i,t}|S_{i-1,t}) + h_t(v_{i,t})$ (in case $v_{i,t} \in S_{i-1,t}$, we repeatedly take samples from the weight distribution of $\mathcal{A}_i$ over the experts until we observe an expert not in the set $S_{i-1,t}$). This technique was first introduced by [22] for the corresponding offline problem and it allows us to obtain $(1 - \frac{c_f}{e})$-regret bounds (where $f = \sum_{t=1}^{T} f_t$) with the optimal approximation ratio (optimality was shown by [23]) instead of the $(1 - \frac{1}{e})$-regret bounds for the online greedy algorithm.

One particular choice for $\{\mathcal{A}_i\}_{i=1}^{m}$ is the WSU algorithm of [10]. We summarize the result for this choice in the theorem below.

**Theorem 6.** *For all $i \in [m]$, let $\mathcal{A}_i$ be an instantiation of the WSU algorithm of [10] for the 1-expert problem and denote $f = \sum_{t=1}^{T} f_t$. The online distorted greedy algorithm applied to the $m$-experts problem obtains the following regret bound:*

$$\mathbb{E}[(1 - \frac{c_f}{e})\text{-}R_T] \leq \sum_{i=1}^{m} R_T^{(i)},$$

*where $R_T^{(i)}$ is the regret of algorithm $\mathcal{A}_i$. If $c_f = 0$, the algorithm is incentive-compatible.*

If $c_f \in (0, 1]$, we can use the online greedy algorithm of [21] instead to ensure incentive compatibility while maintaining similar bounds for the $(1 - \frac{1}{e})$-regret. [10] provided $\mathcal{O}(\sqrt{T \ln K})$ regret bounds for the WSU algorithm. If we plug in this bound in the result of Theorem 6, the regret bound of the online distorted greedy algorithm is $\mathcal{O}(m\sqrt{T \ln K})$. However, this bound depends linearly on $m$ which is suboptimal. To remedy this issue, we first provide an adaptive regret bound for the WSU algorithm below.

**Theorem 7.** *The regret of the WSU algorithm of [10] is bounded by $\mathcal{O}(\sqrt{|L_T| \ln K} + \ln K)$ where $|L_T|$ is the cumulative absolute loss of the algorithm.*

Note that the bound in Theorem 7 adapts to the hardness of the problem. In the worst case, we have $|L_T| = T$ and recover the $\mathcal{O}(\sqrt{T \ln K})$ bound proved in [10]. However, for smaller values of $|L_T|$, the bound in Theorem 7 improves that of [10]. For the non-strategic setting with modular utilities, [7] proposed the Component Hedge (CH) algorithm and obtained a regret bound of $\sqrt{2m\ell^* \ln(\frac{K}{m})} + m \ln(\frac{K}{m})$ where $\ell^*$ is the cumulative loss of the best-chosen set in hindsight. They also gave a matching lower bound for this problem. Applying the same analysis as in Theorem 7 to the setting of the naive approach with $\binom{K}{m}$ meta-experts, we can show that the regret bound of WSU matches the aforementioned lower bound.

We can use the above adaptive regret bound to improve the regret bound of the online distorted greedy algorithm. First, note that at round $t \in [T]$, the sum of the absolute value of losses incurred by $\{\mathcal{A}_i\}_{i=1}^{m}$ is bounded as follows:

$$\sum_{i=1}^{m} \left((1 - \frac{1}{m})^{m-i} g_t(v_{i,t}|S_{i-1,t}) + h_t(v_{i,t})\right) \leq \sum_{i=1}^{m} \underbrace{\left(g_t(v_{i,t}|S_{i-1,t}) + h_t(v_{i,t})\right)}_{= f_t(v_{i,t}|S_{i-1,t})} = f_t(S_t) \leq 1.$$

Therefore, if we denote the cumulative absolute losses incurred by $\mathcal{A}_i$ with $|L_T^{(i)}|$, we have:

$$\sum_{i=1}^{m} |L_T^{(i)}| \leq T.$$

---
**Algorithm 1** Online distorted greedy algorithm
---
**Initialization:** Initialize $m$ instances $\mathcal{A}_1, \ldots, \mathcal{A}_m$ of online algorithms for the 1-expert problem.
**for** $t = 1, \ldots, T$ **do**
    **for** $i = 1, \ldots, m$ **do**
        $\mathcal{A}_i$ chooses the expert $v_{i,t}$ and $S_{i,t} = \{v_{1,t}, \ldots, v_{i,t}\}$.
    **end for**
    Play the set $S_t = S_{m,t} = \{v_{1,t}, \ldots, v_{m,t}\}$ and observe $f_t$.
    Compute the modular function $h_t(S) = \sum_{i \in S} f_t(i|[K] \setminus \{i\})$ and set $g_t = f_t - h_t$.
    **for** $i = 1, \ldots, m$ **do**
        Feedback the cost $-(1 - \frac{1}{m})^{m-i} g_t(v_{i,t}|S_{i-1,t}) - h_t(v_{i,t})$ to $\mathcal{A}_i$.
    **end for**
**end for**
---

Using the result of Theorem 7, we know that the regret bound of the online distorted greedy algorithm is $\sum_{i=1}^{m} \sqrt{|L_T^{(i)}| \ln K} + m \ln K$. $\sum_{i=1}^{m} \sqrt{|L_T^{(i)}|}$ is maximized when $|L_T^{(i)}| = \frac{T}{m}$ for all $i \in [m]$. Thus, in the worst case, the expected $(1 - \frac{c_f}{e})$-regret bound is $\mathcal{O}(\sqrt{mT \ln K} + m \ln K)$.
While we focused on submodular utility functions in this section, we can also apply the online distorted greedy algorithm to the setting with modular utilities. In this case, we have $c_f = 0$, and therefore, the algorithm is incentive-compatible and its 1-regret is bounded by $\mathcal{O}(\sqrt{mT \ln K} + m \ln K)$. Unlike the FTPL algorithm that is only approximately incentive-compatible, the online distorted greedy algorithm applied to modular utility functions is incentive-compatible. However, this comes at the price of an extra $\sqrt{m}$ term in the regret bound.

## 6  Experiments

In this section, we evaluate the performance of our proposed algorithms for modular utility functions on a publicly available dataset from a FiveThirtyEight forecasting competition[4] in which forecasters make predictions about the match outcomes of the 2022–2023 National Football League (NFL). Before each match, FiveThirtyEight provides information on the past performance of the two opposing teams. Forecasters observe this information and make probabilistic predictions about the likelihood of each team winning the match. Considering that there are 284 different matches in the dataset, we set $T = 284$. Out of the 9982 forecasters who participated in this competition, only 274 made predictions for every single match. We consider two cases: $K = 20$ and $K = 100$. To reduce variance, for each case, we sample 5 groups of $K$ forecasters from the 274 and run FTPL and Online Distorted Greedy (ODG) 10 times. We set $m = 5$. Given that FTPL is only approximately incentive-compatible and according to the result of Theorem 3, the reported beliefs could be $\frac{2B}{\eta - 2B}$ distant from the true beliefs, we add a uniformly random value in the range $[\frac{-2B}{\eta - 2B}, \frac{2B}{\eta - 2B}]$ to the true beliefs to model this fact. We use the standard Laplace distribution as the perturbation for FTPL. Hence, we set $B = 1$. For both algorithms, the step size $\eta$ is chosen according to our theoretical results. In Figure 1, we plot the average regret $\frac{1}{t} \mathbb{E}\left[ \max_{S \subseteq [K]:|S|=m} \sum_{\tau=1}^{t} f_\tau(S) - \sum_{\tau=1}^{t} f_\tau(S_\tau) \right]$ of the two algorithms over time (along with error bands corresponding to 20th and 80th percentiles) along with that of the FiveThirtyEight aggregated predictions for $K = 20$ and $K = 100$ settings. Note that while our proposed algorithms choose $m$ predictions at each round $t \in [T]$, the FiveThirtyEight aggregated prediction is a single scalar value. The plots suggest that while the regret of all three algorithms converges to zero as $t$ gets larger, both our proposed algorithms have superior performance compared to that of the FiveThirtyEight predictions.

## 7  Conclusion and future directions

In this paper, we studied the $m$-*experts* problem, a generalization of the standard binary prediction with expert advice problem where at each round $t \in [T]$: 1) The algorithm is allowed to pick $m \geq 1$ experts and its utility is a modular or submodular function of the chosen experts, and 2) The experts are strategic and may misreport their true beliefs about the $t$-th event to increase their probability of

---
[4]https://github.com/fivethirtyeight/nfl-elo-game

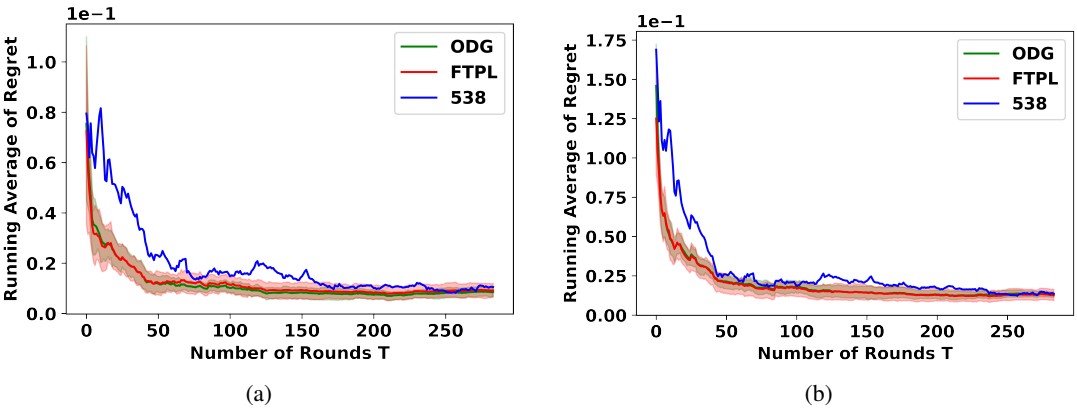

Figure 1: Running average of regret over time for (a) $K = 20$ and (b) $K = 100$.

being chosen at the next round (round $t + 1$). The goal is to design algorithms that incentivize experts to report truthfully (i.e., incentive-compatible) and obtain sublinear regret bounds with respect to the true beliefs of the experts (i.e., no-regret). We proposed two algorithmic frameworks for this problem. In Section 4, we introduced the Follow the Perturbed Leader (FTPL) algorithm. Under a certain condition for the noise distribution (Condition 1), this algorithm is approximately incentive-compatible and achieves sublinear regret bounds for modular utility functions. Moreover, in Section 5, we proposed the online distorted greedy algorithm that applies to both modular and submodular utility functions. This algorithm is incentive-compatible but its regret bound is slightly worse than that of FTPL.

This work could be extended in several interesting directions. First, none of the algorithms discussed here or in prior works have taken into account the properties of the quadratic loss function. In particular, this loss function is exp-concave, and [4] showed that for exp-concave loss functions in the 1-expert problem, the regret bound could be improved to $\mathcal{O}(\ln K)$ using the Hedge algorithm without the incentive compatibility property. Designing incentive-compatible algorithms with similarly improved regret bounds for the 1-expert and $m$-experts problems is yet to be done. In order to obtain the $O(\ln K)$ regret bound for the 1-expert problem (with squared loss) using the Hedge algorithm, the algorithm makes a single prediction $\sum_{i=1}^{K} \pi_{i,t} p_{i,t}$ at round $t \in [T]$ and its loss is $(\sum_{i=1}^{K} \pi_{i,t} p_{i,t} - r_t)^2$. In other words, choosing an expert $i \in [K]$ with probability $\pi_{i,t}$ at round $t$ is not good enough to obtain the improved $\mathcal{O}(\ln K)$ regret bound. Moving on to the $m$-expert problem, the main challenge for obtaining regret bounds better than $\mathcal{O}(\sqrt{T})$ is to decide how to aggregate the $K$ predictions as $m$ scalar values. Second, while we focused on the particular choice of quadratic loss functions, the setting could be extended to other loss functions as well. It is not clear to what extent our results hold when moving beyond the quadratic loss function. Finally, [16] introduced a framework for augmenting online algorithms for various online problems with predictions or pieces of advice. An interesting future research direction is to extend this setting to the case where the predictions are given by strategic experts and study incentive compatibility guarantees for online problems beyond the $m$-experts problem.

## Acknowledgments and Disclosure of Funding

This work was supported in part by the following grants: NSF TRIPODS II 2023166, NSF CIF 2212261, NSF CIF 2007036, NSF AI Inst 2112085.

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

# Appendix

## A  Missing proofs

### A.1  Proof of Theorem 1

At time $t = 1$, we set $\pi_{S,1} = \frac{1}{\binom{K}{m}}$ for each set $S$ where $|S| = m$ and update these weights as follows:

$$\pi_{S,t+1} = \pi_{S,t}(1 - \eta L_{S,t}),$$

where $L_{S,t} = \ell_{S,t} - \sum_{S':|S'|=m} \pi_{S',t}\ell_{S',t}$ and $\ell_{S,t} = \frac{1}{m}\sum_{i \in S} \ell_{i,t}$. If we denote the optimal set as $S^*$, we can write:

$$1 \geq \pi_{S^*,T+1} = \frac{1}{\binom{K}{m}}\prod_{t=1}^{T}(1 - \eta L_{S^*,t}).$$

Taking the natural logarithm of both sides, we have:

$$0 \geq -\ln\binom{K}{m} + \sum_{t=1}^{T}\ln(1 - \eta L_{S^*,t}).$$

We can use the inequality $\ln(1 - x) \geq -x - x^2$ for $x \leq \frac{1}{2}$ (we choose $\eta$ later such that this inequality holds) to obtain:

$$0 \geq -\ln\binom{K}{m} - \eta\sum_{t=1}^{T} L_{S^*,t} - \eta^2\sum_{t=1}^{T} L_{S^*,t}^2.$$

Rearranging the terms and dividing both sides by $\eta$, we have:

$$-\sum_{t=1}^{T} L_{S^*,t} \leq \frac{\ln\binom{K}{m}}{\eta} + \eta\sum_{t=1}^{T} L_{S^*,t}^2.$$

Using the fact that $R_T = -\sum_{t=1}^{T} L_{S^*,t}$, the inequality $\binom{K}{m} \leq (\frac{Ke}{m})^m$ and $|L_{S,t}| \leq 1\ \forall S, t$, we can write:

$$R_T \leq \frac{m\ln(\frac{Ke}{m})}{\eta} + \eta T.$$

Setting $\eta = \sqrt{\frac{m\ln(\frac{Ke}{m})}{T}}$, we obtain the regret bound $\mathcal{O}(\sqrt{mT\ln(\frac{K}{m})})$. Note that the assumption $\eta L_{S^*,t} \leq \frac{1}{2}$ (used in the proof) holds if $T \geq 4m\ln(\frac{Ke}{m})$. Therefore, we assume $T$ is large enough to satisfy this inequality.

### A.2  Proof of Theorem 2

We can show that if $B = \max_z |\nu'(z)|$, the hazard rate of the distribution is bounded above by $B$ as well. To see this, fix $x \geq 0$. Note that $\frac{f(-x)}{1-F(-x)} = \frac{f(x)}{1-F(-x)} \leq \frac{f(x)}{1-F(x)}$ for a symmetric zero-mean distribution and therefore, we only need to bound the hazard rate at $x > 0$ to bound $\text{haz}_{\mathcal{D}}$. We can

write:

$$\frac{f_{\mathcal{D}}(x)}{1 - F_{\mathcal{D}}(x)} = \frac{f_{\mathcal{D}}(x)}{\int_x^\infty f_{\mathcal{D}}(z)dz}$$

$$= \frac{\exp(-\nu(x))}{\int_x^\infty \exp(-\nu(z))dz}$$

$$= \frac{1}{\int_x^\infty \exp(\nu(x) - \nu(z))dz}$$

$$\stackrel{(a)}{=} \frac{1}{\int_x^\infty \exp((x - z)\nu'(z_x))dz}$$

$$\stackrel{(b)}{\leq} \frac{1}{\int_x^\infty \exp(B(x - z))dz}$$

$$= \frac{1}{(1/B)}$$

$$= B,$$

where we have used the mean-value theorem in (a) and $z_x$ is in the line segment between $x$ and $z$. Also, note that since $x, z > 0$, $z_x > 0$ holds as well and therefore, $\nu'(z_x) > 0$. We have used this fact along with $x - z < 0$ to obtain (b). Therefore, we have $\mathrm{haz}_{\mathcal{D}} \leq B$.

### A.3  Proof of Theorem 3

Let's fix round $t \in [T]$ and expert $i \in [K]$. For $j \neq i$, denote $x_{j,0}^{(t)} = \sum_{s=1}^{t-1} \ell_{j,s} + p_{j,t}^2 + \eta\gamma_{j,t}$ as the total losses of expert $j$ up to round $t$ plus noise if $r_t = 0$. Similarly, we can define $x_{j,1}^{(t)} = \sum_{s=1}^{t-1} \ell_{j,s} + (1 - p_{j,t})^2 + \eta\gamma_{j,t}$. Define $X_0^{(t)}$ ($X_1^{(t)}$) as the $m$-th smallest value in $\{x_{j,0}^{(t)}\}_{j\neq i}$ ($\{x_{j,1}^{(t)}\}_{j\neq i}$). Note that $|X_0^{(t)} - X_1^{(t)}| \leq 1$ holds because for each $j \neq i$, we have $|x_{j,0}^{(t)} - x_{j,1}^{(t)}| \leq 1$. Also, let $L = \sum_{s=1}^{t-1} \ell_{i,s}$. If $r_t = 0$, expert $i$ is chosen at round $t + 1$ if and only if $L + p_{i,t}^2 + \eta\gamma_{i,t} <= X_0^{(t)}$. Similarly, for the case $r_t = 1$, expert $i$ is chosen if and only if $L + (1 - p_{i,t})^2 + \eta\gamma_{i,t} <= X_1^{(t)}$. Rearranging the terms, we can write:

$$\eta\gamma_{i,t} + L - X_0^{(t)} \leq -p_{i,t}^2 \quad \text{if } r_t = 0,$$

$$\eta\gamma_{i,t} + L - X_1^{(t)} \leq -(1 - p_{i,t})^2 \quad \text{if } r_t = 1.$$

Given that $f(\gamma_{i,t}) \propto \exp(-\nu(\gamma_{i,t}))$, if we define $Y_0 = \eta\gamma_{i,t} + L - X_0^{(t)}$ and $Y_1 = \eta\gamma_{i,t} + L - X_1^{(t)}$, we have:

$$f_0(Y_0) \propto \exp(-\nu(\frac{Y_0 - (L - X_0^{(t)})}{\eta})),$$

$$f_1(Y_1) \propto \exp(-\nu(\frac{Y_1 - (L - X_1^{(t)})}{\eta})).$$

Therefore, if $F_0$ and $F_1$ denote the cdf, we can write the probability of expert $i$ being chosen at round $t + 1$ as $F_0(-p_{i,t}^2)$ and $F_1(-(1 - p_{i,t})^2)$ for the cases $r_t = 0$ and $r_t = 1$ respectively. Putting the above results together, we can write the expected utility of expert $i$ (according to her belief $b_{i,t}$) at round $t$ as follows:

$$\mathbb{E}_{r_t \sim \text{Bernoulli}(b_{i,t})}[U_{i,t}] = (1 - b_{i,t})F_0(-p_{i,t}^2) + b_{i,t}F_1(-(1 - p_{i,t})^2).$$

Taking the derivative and setting it to zero, we have:

$$\frac{d}{dp_{i,t}}\mathbb{E}_{r_t \sim \text{Bernoulli}(b_{i,t})}[U_{i,t}] = -2p_{i,t}(1-b_{i,t})f_0(-p_{i,t}^2) + 2(1-p_{i,t})b_{i,t}f_1(-(1-p_{i,t})^2) = 0,$$

$$2f_1(-(1-p_{i,t})^2)\Big(-p_{i,t}(1-b_{i,t})\frac{f_0(-p_{i,t}^2)}{f_1(-(1-p_{i,t})^2)} + (1-p_{i,t})b_{i,t}\Big) = 0,$$

$$-p_{i,t}(1-b_{i,t})\frac{\exp(-\nu(\frac{-p_{i,t}^2-(L-X_0^{(t)})}{\eta}))}{\exp(-\nu(\frac{-(1-p_{i,t})^2-(L-X_1^{(t)})}{\eta}))} + (1-p_{i,t})b_{i,t} = 0,$$

$$-p_{i,t}(1-b_{i,t})\underbrace{\exp\Big(\nu(\frac{-(1-p_{i,t})^2-(L-X_1^{(t)})}{\eta}) - \nu(\frac{-p_{i,t}^2-(L-X_0^{(t)})}{\eta})\Big)}_{=A} + (1-p_{i,t})b_{i,t} = 0.$$

Ideally, we want $A$ to be as close to 1 as possible because if $A = 1$, $p_{i,t}^* = b_{i,t}$ and the algorithm would be incentive-compatible. In general, we have:

$$p_{i,t}^* = \frac{b_{i,t}}{b_{i,t} + (1-b_{i,t})A}.$$

Note that this is not a closed-form solution for $p_{i,t}^*$ because $A$ is also a function of $p_{i,t}$. We can observe that for $p_{i,t} < p_{i,t}^*$, the derivative of the utility function is positive, and for $p_{i,t} > p_{i,t}^*$, the derivative is negative.

We can bound A as follows: Let $g(u) = f(\frac{-p_{i,t}^2-a-u(a'-a)}{\eta})$ where $a = L - X_1^{(t)} + 1 - 2p_{i,t}$, $a' = L - X_0^{(t)}$ and $f(z) = \exp(-\nu(z))$. Taking derivative of $g$ with respect to $u$, we obtain:

$$g'(u) = \frac{(a'-a)\nu'(\frac{-p_{i,t}^2-a-u(a'-a)}{\eta})}{\eta}f(\frac{-p_{i,t}^2-a-u(a'-a)}{\eta}) = \frac{(a'-a)\nu'(\frac{-p_{i,t}^2-a-u(a'-a)}{\eta})}{\eta}g(u).$$

Therefore, we can write:

$$\ln f(\frac{-p_{i,t}^2-a'}{\eta}) - \ln f(\frac{-p_{i,t}^2-a}{\eta}) = \ln g(1) - \ln g(0)$$

$$= \int_0^1 \frac{g'(u)}{g(u)}du$$

$$= \int_0^1 \frac{(a'-a)\nu'(\frac{-p_{i,t}^2-a-u(a'-a)}{\eta})}{\eta}du.$$

We have $a' - a = X_1^{(t)} - X_0^{(t)} + 2p_{i,t} - 1$. Therefore, $-2 \leq a' - a \leq 2$ holds. Moreover, we have $-B \leq \nu'(\frac{-p_{i,t}^2-a-u(a'-a)}{\eta}) \leq B$ due to Condition 1. Putting the above results together, the following holds:

$$\frac{-2B}{\eta} \leq \ln A = \ln f(\frac{-p_{i,t}^2-a'}{\eta}) - \ln f(\frac{-p_{i,t}^2-a}{\eta}) \leq \frac{2B}{\eta}.$$

Therefore, $A \in [\exp(\frac{-2B}{\eta}), \exp(\frac{2B}{\eta})]$. Let $h(p_{i,t}) = p_{i,t} - p_{i,t}^* = p_{i,t} - \frac{b_{i,t}}{b_{i,t}+(1-b_{i,t})A}$. Since $p_{i,t}^* \in [0,1]$, we have $h(0) < 0$ and $h(1) > 0$. Taking the derivative of $h$ with respect to $p_{i,t}$, we have:

$$\frac{dh}{dp_{i,t}} = 1 + \frac{b_{i,t}(1-b_{i,t})A\Big(\frac{2(1-p_{i,t})}{\eta}\nu'(\frac{-(1-p_{i,t})^2-(L-X_1^{(t)})}{\eta}) + \frac{2p_{i,t}}{\eta}\nu'(\frac{-p_{i,t}^2-(L-X_0^{(t)})}{\eta})\Big)}{(b_{i,t}+(1-b_{i,t})A)^2}.$$

Using Condition 1, we have $1 - \frac{2Bb_{i,t}(1-b_{i,t})A}{\eta(b_{i,t}+(1-b_{i,t})A)^2} \leq \frac{dh}{dp_{i,t}}$. Therefore, we can write:

$$\frac{dh}{dp_{i,t}} \geq 1 - \frac{2Bb_{i,t}(1-b_{i,t})A}{\eta(b_{i,t}+(1-b_{i,t})A)^2} = 1 - \frac{2Bb_{i,t}(1-b_{i,t})A}{\eta(b_{i,t}^2+(1-b_{i,t})^2A^2+2b_{i,t}(1-b_{i,t})A)} \geq 1 - \frac{2Bb_{i,t}(1-b_{i,t})A}{2\eta b_{i,t}(1-b_{i,t})A} = 1 - \frac{B}{\eta} > 0.$$

We choose $\eta$ later such that $\eta > B$ for the last inequality to hold. So, $h$ is strictly increasing, there is exactly one solution $p_{i,t}^*$, and the derivative is positive below it and negative above it.

If we replace $A$ with something larger in $p_{i,t}^*$, the value decreases and vice versa. Therefore, we can write:

$$
\begin{aligned}
p_{i,t}^* &\leq \frac{b_{i,t}}{b_{i,t} + (1 - b_{i,t})\exp(\frac{-2B}{\eta})} \\
&= \frac{b_{i,t}}{\exp(\frac{-2B}{\eta}) + b_{i,t}(1 - \exp(\frac{-2B}{\eta}))} \\
&\leq \frac{b_{i,t}}{\exp(\frac{-2B}{\eta})} \\
&\leq \frac{b_{i,t}}{1 - \frac{2B}{\eta}} \\
&= \frac{\eta b_{i,t}}{\eta - 2B} \\
&= b_{i,t} + \frac{2B b_{i,t}}{\eta - 2B} \\
&\leq b_{i,t} + \frac{2B}{\eta - 2B}.
\end{aligned}
$$

Similarly, we can lower bound $p_{i,t}^*$ as follows:

$$
\begin{aligned}
p_{i,t}^* &\geq \frac{b_{i,t}}{b_{i,t} + (1 - b_{i,t})\exp(\frac{2B}{\eta})} \\
&= \frac{b_{i,t}}{\exp(\frac{2B}{\eta}) - b_{i,t}(\exp(\frac{2B}{\eta}) - 1)} \\
&\geq \frac{b_{i,t}}{\exp(\frac{2B}{\eta})} \\
&= b_{i,t}\exp(\frac{-2B}{\eta}) \\
&\geq b_{i,t}(1 - \frac{2B}{\eta}) \\
&\geq b_{i,t} - \frac{2B}{\eta}.
\end{aligned}
$$

Putting the above results together, we conclude that $|p_{i,t}^* - b_{i,t}| \leq \frac{2B}{\eta - 2B}$ holds.

## A.4 Proof of Theorem 4

Let $\eta > 0$ and $\mathcal{X}$ be the set of $\binom{K}{m}$ feasible sets of size $m$ for this problem. Denote $L_t \in \mathbb{R}^K$ as the partial sum of losses for all experts before round $t$, i.e., $[L_t]_i = \sum_{s=1}^{t-1} \ell_{i,s}$. The update rule of FTPL at round $t \in [T]$ is $\pi_t = \arg\min_{x \in \mathcal{X}} \langle x, L_t + \eta\gamma_t \rangle$. First, we analyze the expected regret of the algorithm below.

Define $\phi_t(\theta) = \mathbb{E}_{\gamma_t \sim \mathcal{D}^K}[\min_{x \in \mathcal{X}} \langle x, \theta + \eta\gamma_t \rangle]$. Then, we have $\nabla\phi_t(L_t) = \mathbb{E}_{\gamma_t \sim \mathcal{D}^K}[\arg\min_{x \in \mathcal{X}} \langle x, L_t + \eta\gamma_t \rangle] = \mathbb{E}_{\gamma_t \sim \mathcal{D}^K}[\pi_t]$ and $\langle \nabla\phi_t(L_t), \ell_t \rangle = \mathbb{E}_{\gamma_t \sim \mathcal{D}^K}[\langle \pi_t, \ell_t \rangle]$. Using the Taylor's expansion of $\phi_t$, we can write:

$$
\phi_t(L_{t+1}) = \phi_t(L_t) + \langle \nabla\phi_t(L_t), \ell_t \rangle + \frac{1}{2}\langle \ell_t, \nabla^2\phi_t(L_t')\ell_t \rangle = \phi_t(L_t) + \mathbb{E}_{\gamma_t \sim \mathcal{D}^K}[\langle \pi_t, \ell_t \rangle] + \frac{1}{2}\langle \ell_t, \nabla^2\phi_t(L_t')\ell_t \rangle,
$$

where $L_t'$ is in the line segment between $L_t$ and $L_{t+1} = L_t + \ell_t$. By definition, $\phi_t$ is the minimum of linear functions and therefore, it is concave. Thus, $\nabla^2\phi_t(L_t')$ is negative semidefinite. Note that since $\gamma_t$ is simply $K$ i.i.d. samples of the noise distribution for all $t \in [T]$, we have $\phi_t(\theta) = \phi(\theta) = $

$\mathbb{E}_{\gamma \sim \mathcal{D}^K}[\min_{x \in \mathcal{X}} \langle x, \theta + \eta \gamma \rangle]$. Taking the sum over $t \in [T]$ and rearranging the terms, we obtain:

$$\mathbb{E}[\sum_{t=1}^{T} \langle \pi_t, \ell_t \rangle] = \phi(L_{T+1}) - \phi(\underbrace{L_1}_{=0}) - \frac{1}{2} \sum_{t=1}^{T} \langle \ell_t, \nabla^2 \phi_t(L_t^{'}) \ell_t \rangle.$$

Using Jensen's inequality, we can write:

$$\phi(L_{T+1}) = \mathbb{E}_{\gamma \sim \mathcal{D}^K}[\min_{x \in \mathcal{X}} \langle x, L_{T+1} + \eta \gamma \rangle] \leq \min_{x \in \mathcal{X}} \mathbb{E}_{\gamma \sim \mathcal{D}^K}[\langle x, L_{T+1} + \eta \gamma \rangle] = \min_{x \in \mathcal{X}} \langle x, L_{T+1} \rangle.$$

Therefore, we can rearrange the terms and bound the expected regret as follows:

$$\mathbb{E}[R_T] = \frac{1}{m} \mathbb{E}_{\gamma \sim \mathcal{D}^K}[\sum_{t=1}^{T} \langle \pi_t, \ell_t \rangle] - \frac{1}{m} \min_{x \in \mathcal{X}} \langle x, L_{T+1} \rangle \leq -\frac{1}{m} \phi(0) - \frac{1}{2m} \sum_{t=1}^{T} \langle \ell_t, \nabla^2 \phi_t(L_t^{'}) \ell_t \rangle.$$

To bound the first term on the right hand side, we can use the fact that $\mathcal{D}$ is symmetric to write:

$$-\phi(0) = -\mathbb{E}_{\gamma \sim \mathcal{D}^K}[\min_{x \in \mathcal{X}} \langle x, \eta \gamma \rangle] = \eta \mathbb{E}_{\gamma \sim \mathcal{D}^K}[\max_{x \in \mathcal{X}} \langle x, -\gamma \rangle] = \eta \mathbb{E}_{\gamma \sim \mathcal{D}^K}[\max_{x \in \mathcal{X}} \langle x, \gamma \rangle].$$

Now, we move on to bound the second term in the regret bound. Given that the losses of the experts at each round $t \in [T]$ are bounded between 0 and 1, we have $\|\ell_t\|_\infty \leq 1$. Therefore, we have:

$$-\langle \ell_t, \nabla^2 \phi_t(L_t^{'}) \ell_t \rangle \leq \sum_{i,j \in [K]} |\nabla_{i,j}^2 \phi_t(L_t^{'})|.$$

By definition, if we denote $\hat{\pi}(\theta) = \arg\min_{x \in \mathcal{X}} \langle x, \theta \rangle$, we have:

$$\nabla_{i,j}^2 \phi(L_t^{'}) = \frac{1}{\eta} \mathbb{E}_{\gamma \sim \mathcal{D}^K}[\hat{\pi}(L_t^{'} + \eta \gamma)_i \frac{d\nu(\gamma_j)}{d\gamma_j}].$$

Given the concavity of $\phi$, diagonal entries of the Hessian $\nabla^2 \phi$ are non-positive. We can also use $1^T \hat{\pi}(\theta) = m$ to show that the off-diagonal entries are non-negative and each row or column of the Hessian sums up to 0. Therefore, we have $\sum_{i,j \in [K]} |\nabla_{i,j}^2 \phi(L_t^{'})| = -2 \sum_{i=1}^{K} \nabla_{i,i}^2 \phi(L_t^{'})$. Putting the above results together, we can bound the regret as follows:

$$\mathbb{E}[R_T] \leq \frac{\eta}{m} \mathbb{E}_{\gamma \sim \mathcal{D}^K}[\max_{x \in \mathcal{X}} \langle x, \gamma \rangle] + \frac{1}{\eta m} \sum_{t=1}^{T} \sum_{i=1}^{K} \mathbb{E}_{\gamma \sim \mathcal{D}^K}[\hat{\pi}(L_t^{'} - \eta \gamma)_i \frac{d\nu(\gamma_i)}{d\gamma_i}].$$

We can bound the first term as follows:

$$\mathbb{E}_{\gamma \sim \mathcal{D}^K}[\max_{x \in \mathcal{X}} \langle x, \gamma \rangle] \leq \inf_{s:s>0} \frac{1}{s} \ln \big( \sum_{x \in \mathcal{X}} \mathbb{E}_{\gamma \sim \mathcal{D}^K}[\exp(s \langle x, \gamma \rangle)] \big)$$

$$= \inf_{s:s>0} \frac{1}{s} \ln \big( \sum_{x \in \mathcal{X}} \prod_{i=1}^{K} \mathbb{E}_{\gamma_i \sim \mathcal{D}}[\exp(s x_i \gamma_i)] \big)$$

$$= \inf_{s:s>0} \frac{1}{s} \ln \big( |\mathcal{X}| (\mathbb{E}_{\gamma_1 \sim \mathcal{D}}[\exp(s \gamma_1)])^m \big)$$

$$= \inf_{s:s>0} \frac{1}{s} \ln |\mathcal{X}| + \frac{m}{s} \ln \mathbb{E}_{\gamma_1 \sim \mathcal{D}}[\exp(s \gamma_1)]$$

$$\leq \inf_{s:s>0} \frac{m}{s} \ln(\frac{Ke}{m}) + \frac{m}{s} \ln \mathbb{E}_{\gamma_1 \sim \mathcal{D}}[\exp(s \gamma_1)]$$

$$\leq \mathcal{O}(m \ln(\frac{K}{m})) + \mathcal{O}(m)$$

$$= \mathcal{O}(m \ln(\frac{K}{m})).$$

To bound the second term in the regret bound, we can use Condition 1 to write:

$$\sum_{i=1}^{K} \mathbb{E}_{\gamma \sim \mathcal{D}}[\hat{\pi}(L_t^{'} - \eta \gamma)_i \frac{d\nu(\gamma_i)}{d\gamma_i}] \leq B \sum_{i=1}^{K} \mathbb{E}_{\gamma \sim \mathcal{D}}[\hat{\pi}(L_t^{'} - \eta \gamma)_i] = B \mathbb{E}_{\gamma \sim \mathcal{D}}[\sum_{i=1}^{K} \hat{\pi}(L_t^{'} - \eta \gamma)_i] = Bm.$$

Putting the above results together, we have:

$$\mathbb{E}[R_T] \leq \mathcal{O}(\eta \ln(\frac{K}{m})) + \frac{BT}{\eta}.$$

Therefore, if we set $\eta = \sqrt{\frac{BT}{\ln(\frac{K}{m})}}$, the regret bound is $\mathcal{O}(\sqrt{BT \ln(\frac{K}{m})})$. In the proof of Theorem 3, we assumed that $\eta$ is chosen such that $\eta > B$. So, we assume $T > B \ln(\frac{K}{m})$.

### A.5 Proof of Theorem 6

For $i \in [m]$ and $t \in [T]$, define:

$$\phi_{i,t}(S) = (1 - \frac{1}{m})^{m-i} g_t(S) + h_t(S).$$

For all $i \in [m]$, we can write

$$\phi_{i,t}(S_{i,t}) - \phi_{i-1,t}(S_{i-1,t}) = (1 - \frac{1}{m})^{m-i} g_t(S_{i,t}) + h_t(S_{i,t}) - (1 - \frac{1}{m})^{m-(i-1)} g_t(S_{i-1,t}) - h_t(S_{i-1,t})$$

$$= (1 - \frac{1}{m})^{m-i} (g_t(S_{i,t}) - g_t(S_{i-1,t})) + h_t(v_{i,t}) + \frac{1}{m}(1 - \frac{1}{m})^{m-i} g_t(S_{i-1,t}).$$

Taking the sum over $t \in [T]$, we obtain:

$$\sum_{t=1}^{T}(\phi_{i,t}(S_{i,t}) - \phi_{i-1,t}(S_{i-1,t})) = (1 - \frac{1}{m})^{m-i} \sum_{t=1}^{T}(g_t(S_{i,t}) - g_t(S_{i-1,t})) + \sum_{t=1}^{T} h_t(v_{i,t}) + \frac{1}{m}(1 - \frac{1}{m})^{m-i} \sum_{t=1}^{T} g_t(S_{i-1,t})$$

$$= (1 - \frac{1}{m})^{m-i} \sum_{t=1}^{T} g_t(v_{i,t}|S_{i-1,t}) + \sum_{t=1}^{T} h_t(v_{i,t}) + \frac{1}{m}(1 - \frac{1}{m})^{m-i} \sum_{t=1}^{T} g_t(S_{i-1,t}).$$

If the regret of $\mathcal{A}_i$ is bounded above by $R_T^{(i)}$ and the optimal benchmark solution is OPT $= \{v_1^*, \ldots, v_m^*\}$, for all $j \in [m]$, we can write:

$$\sum_{t=1}^{T}\left((1 - \frac{1}{m})^{m-i} g_t(v_j^*|S_{i-1,t}) + h_t(v_j^*)\right) - \sum_{t=1}^{T}\left((1 - \frac{1}{m})^{m-i} g_t(v_{i,t}|S_{i-1,t}) + h_t(v_{i,t})\right) \leq R_T^{(i)}.$$

Putting the above inequalities together, we have:

$$\sum_{t=1}^{T}(\phi_{i,t}(S_{i,t}) - \phi_{i-1,t}(S_{i-1,t})) \geq \frac{1}{m}(1 - \frac{1}{m})^{m-i} \sum_{t=1}^{T}\sum_{j=1}^{m} g_t(v_j^*|S_{i-1,t}) + \frac{1}{m}\sum_{t=1}^{T}\sum_{j=1}^{m} h_t(v_j^*)$$

$$+ \frac{1}{m}(1 - \frac{1}{m})^{m-i} \sum_{t=1}^{T} g_t(S_{i-1,t}) - R_T^{(i)}.$$

We can use submodularity and monotonicity of $g_t$ to write:

$$g_t(\text{OPT}) - g_t(S_{i-1,t}) \leq g_t(\text{OPT} \cup S_{i-1,t}) - g_t(S_{i-1,t}) = \sum_{j=1}^{m} g_t(v_j^*|S_{i-1,t} \cup \{v_1^*, \ldots, v_{j-1}^*\}) \leq \sum_{j=1}^{m} g_t(v_j^*|S_{i-1,t}).$$

We can combine the last two inequalities to write:

$$\sum_{t=1}^{T}(\phi_{i,t}(S_{i,t}) - \phi_{i-1,t}(S_{i-1,t})) \geq \frac{1}{m}(1 - \frac{1}{m})^{m-i} \sum_{t=1}^{T}(g_t(\text{OPT}) - g_t(S_{i-1,t})) + \frac{1}{m}\sum_{t=1}^{T}\sum_{j=1}^{m} h_t(v_j^*)$$

$$+ \frac{1}{m}(1 - \frac{1}{m})^{m-i} \sum_{t=1}^{T} g_t(S_{i-1,t}) - R_T^{(i)}$$

$$= \frac{1}{m}(1 - \frac{1}{m})^{m-i} \sum_{t=1}^{T} g_t(\text{OPT}) + \frac{1}{m}\sum_{t=1}^{T}\sum_{j=1}^{m} h_t(v_j^*) - R_T^{(i)}.$$

Taking the sum over $i \in [m]$, we have:

$$\sum_{t=1}^{T}\sum_{i=1}^{m}(\phi_{i,t}(S_{i,t}) - \phi_{i-1,t}(S_{i-1,t})) \geq \frac{1}{m}\sum_{i=1}^{m}(1 - \frac{1}{m})^{m-i} \sum_{t=1}^{T} g_t(\text{OPT}) + \sum_{t=1}^{T}\sum_{j=1}^{m} h_t(v_j^*) - \sum_{i=1}^{m} R_T^{(i)}$$

$$= (1 - (1 - \frac{1}{m})^m) \sum_{t=1}^{T} g_t(\text{OPT}) + \sum_{t=1}^{T} h_t(\text{OPT}) - \sum_{i=1}^{m} R_T^{(i)}$$

$$\geq (1 - \frac{1}{e}) \sum_{t=1}^{T} g_t(\text{OPT}) + \sum_{t=1}^{T} h_t(\text{OPT}) - \sum_{i=1}^{m} R_T^{(i)}.$$

On the other hand, we have:

$$\sum_{t=1}^{T}\sum_{i=1}^{m}(\phi_{i,t}(S_{i,t})-\phi_{i-1,t}(S_{i-1,t})) = \sum_{t=1}^{T}(\phi_{m,t}(S_{m,t})-\phi_{0,t}(S_{0,t})) = \sum_{t=1}^{T}(g_t(S_t)+h_t(S_t)) = \sum_{t=1}^{T}f_t(S_t).$$

Combining the last two inequalities, we obtain:

$$\sum_{t=1}^{T}f_t(S_t) \geq (1-\frac{1}{e})\sum_{t=1}^{T}g_t(\text{OPT}) + \sum_{t=1}^{T}h_t(\text{OPT}) - \sum_{i=1}^{m}R_T^{(i)}$$

$$= (1-\frac{1}{e})\sum_{t=1}^{T}f_t(\text{OPT}) - (1-\frac{1}{e})\sum_{t=1}^{T}h_t(\text{OPT}) + \sum_{t=1}^{T}h_t(\text{OPT}) - \sum_{i=1}^{m}R_T^{(i)}$$

$$= (1-\frac{1}{e})\sum_{t=1}^{T}f_t(\text{OPT}) + \frac{1}{e}\sum_{t=1}^{T}h_t(\text{OPT}) - \sum_{i=1}^{m}R_T^{(i)}$$

$$\geq (1-\frac{1}{e})\sum_{t=1}^{T}f_t(\text{OPT}) + \frac{1-c_f}{e}\sum_{t=1}^{T}f_t(\text{OPT}) - \sum_{i=1}^{m}R_T^{(i)}$$

$$= (1-\frac{c_f}{e})\sum_{t=1}^{T}f_t(\text{OPT}) - \sum_{i=1}^{m}R_T^{(i)},$$

where $f = \sum_{t=1}^{T}f_t$. Rearranging the terms, we obtain the $(1-\frac{c_f}{e})$-regret bound of the online distorted greedy algorithm as follows:

$$(1-\frac{c_f}{e})\sum_{t=1}^{T}f_t(\text{OPT}) - \sum_{t=1}^{T}f_t(S_t) \leq \sum_{i=1}^{m}R_T^{(i)}.$$

To see why the online greedy algorithm is incentive-compatible, note that at round $t \in [T]$, the loss of expert $j \in [K]$ for the instance $\mathcal{A}_i; \ i \in [K]$ is:

$$-f_t(j|S_{i-1,t}) = \prod_{k \in S_{i-1,t}} \ell_{k,t}(\ell_{j,t}-1).$$

If we use $\pi_{j,t+1}^{(i)}$ to denote the probability of expert $j$ being chosen at round $t+1$ by $\mathcal{A}_i$, we can write:

$$\pi_{j,t+1}^{(i)} = \pi_{j,t}^{(i)}\big(1-\eta \prod_{k \in S_{i-1,t}} \ell_{k,t}(\ell_{j,t} - \sum_{s=1}^{K}\pi_{s,t}^{(i)}\ell_{s,t})\big).$$

$\pi_{j,t+1}^{(i)}$ is linear in $\ell_{j,t}$ and expert $j$ does not have control over the term $\prod_{k \in S_{i-1,t}} \ell_{k,t}$ (because $S_{i-1,t}$ is unknown to the expert). Therefore, to maximize $\pi_{j,t+1}^{(i)}$, expert $j$ can only aim to minimize $\mathbb{E}_{r_t \sim \text{Bern}(b_{j,t})}[\ell_{j,t} - \sum_{s=1}^{K}\pi_{s,t}^{(i)}\ell_{s,t}]$. Given that the quadratic loss function is proper, we can conclude that $\mathcal{A}_i$ is incentive-compatible. Moreover, since the online greedy algorithm simply outputs the predictions of $\{\mathcal{A}_i\}_{i=1}^{K}$, this algorithm is incentive-compatible as well. For the case with $c_f = 0$, the loss of expert $j \in [K]$ for each instance $\mathcal{A}_i; \ i \in [K]$ in both online greedy algorithm and online distorted greedy algorithm is simply $\frac{1}{m}(\ell_{j,t}-1)$ and the same argument holds.

### A.6 Proof of Theorem 7

First, assume that the losses are non-negative. Using the update rule of the algorithm, we can write:

$$1 \geq \pi_{i^*,T+1} = \pi_{i^*,1}\prod_{t=1}^{T}(1-\eta L_{i^*,t}),$$

$$1 \geq \frac{1}{K}\prod_{t:L_{i^*,t}\geq 0}(1-\eta L_{i^*,t})\prod_{t:L_{i^*,t}<0}(1-\eta L_{i^*,t}).$$

We can use the inequalities $1 - \eta x \geq (1 - \eta)^x$ for $x \in [0, 1]$ and $1 - \eta x \geq (1 + \eta)^{-x}$ for $x \in [-1, 0]$ to write:

$$1 \geq \frac{1}{K} \prod_{t:L_{i^*,t} \geq 0} (1 - \eta)^{L_{i^*,t}} \cdot \prod_{t:L_{i^*,t} < 0} (1 + \eta)^{-L_{i^*,t}},$$

$$0 \geq -\ln K + \sum_{t:L_{i^*,t} \geq 0} L_{i^*,t} \ln(1 - \eta) - \sum_{t:L_{i^*,t} < 0} L_{i^*,t} \ln(1 + \eta).$$

Given the inequalities $\ln(1 - x) \geq -x - x^2$ and $\ln(1 + x) \geq x - x^2$ for $x \leq \frac{1}{2}$, we have:

$$0 \geq -\ln K + (-\eta - \eta^2) \sum_{t:L_{i^*,t} \geq 0} L_{i^*,t} - (\eta - \eta^2) \sum_{t:L_{i^*,t} < 0} L_{i^*,t},$$

$$0 \geq -\ln K - \eta \sum_{t=1}^{T} L_{i^*,t} - \eta^2 \sum_{t=1}^{T} |L_{i^*,t}|,$$

$$R_T = -\sum_{t=1}^{T} L_{i^*,t} \leq \frac{\ln K}{\eta} + \eta \sum_{t=1}^{T} |L_{i^*,t}|.$$

Given that $L_{i^*,t} = \ell_{i^*,t} - \pi_t^T \ell_t$, we can write:

$$R_T \leq \frac{\ln K}{\eta} + \eta \sum_{t=1}^{T} |\ell_{i^*,t} - \pi_t^T \ell_t| \leq \frac{\ln K}{\eta} + \eta \left( \sum_{t=1}^{T} |\ell_{i^*,t}| + \sum_{t=1}^{T} |\pi_t^T \ell_t| \right) = \frac{\ln K}{\eta} + \eta(|L_T^*| + |L_T|),$$

where $|L_T^*|$ and $|L_T|$ are the cumulative absolute loss of the benchmark and the algorithm respectively. Therefore, if we set $\eta = \min\{1, \sqrt{\frac{\ln K}{|L_T| + |L_T^*|}}\}$, we obtain an $\mathcal{O}(\sqrt{\max\{|L_T|, |L_T^*|\} \ln K} + \ln K)$ regret bound. Given that $R_T = L_T - L_T^*$, if $|L_T| = L_T < L_T^* = |L_T^*|$, the regret is negative and if $|L_T| = L_T \geq L_T^* = |L_T^*|$, the regret bound is $\mathcal{O}(\sqrt{|L_T| \ln K} + \ln K)$. Therefore, the regret bound $\mathcal{O}(\sqrt{|L_T| \ln K} + \ln K)$ always holds.

