# OpenReview forum: "No-Regret Online Prediction with Strategic Experts"
_NeurIPS.cc/2023/Conference — NeurIPS 2023 poster_

### Official Review · Reviewer_oxKD · 2023-07-02

**Soundness:** 3 good
**Presentation:** 4 excellent
**Contribution:** 3 good
**Rating:** 6
**Confidence:** 3

**Summary:**

This submission studies the problem of online decision-making using predictions given by experts with incentives to behave strategically. In particular, at each time-step, K experts hold a belief about a binary outcome. Each expert reports a prediction to a learner, who picks a set of m experts. The learner then suffers a loss which is a function of both the binary outcome and the chosen experts' true beliefs about the outcome. The goal of the learner is to achieve no-regret with respect to the best-in-hindsight choice of m experts while maintaining incentive compatibility, i.e. reporting their true belief as their prediction is a weakly-dominant strategy for each expert at every time-step.

The authors study two settings: one in which the learner has a modular utility function and one in which their utility function is submodular. An existing algorithm (WSU, [6]) for the 1-expert version of this problem (i.e. the version of the problem where the learner selects one expert at each time-step) is capable of getting sublinear regret in both settings, albeit at the cost of exponential runtime. For modular utility functions, authors show how to use a variant of the well-known Follow the Perturbed Leader (FTPL) algorithm under a sufficient condition for the perturbation distribution to guarantee both no-regret and approximate incentive compatibility.

For submodular utility functions, the authors use an "online distorted greedy algorithm" to obtain no-regret while maintaining incentive compatibility constraints exactly. At a high level, the online distorted greedy algorithm runs m algorithms for the strategic 1-expert problem concurrently. At each time-step, each of the m sub-algorithms selects an expert, and the learner uses these m experts as their selection. Based on the loss the learner receives, they set the loss of each sub-algorithm in a particular way. The authors use the WSU algorithm of [6] as their sub-algorithm. Along they way, they derive an adaptive regret bound for the WSU algorithm, which may be of independent interest.

Finally, the authors empirically evaluate their two algorithms on a dataset from a FiveThrityEight forecasting competition to predict the outcomes of games in the 2022-2023 NFL football season. They find that both algorithms obtain similar performance in this setting.

**Strengths:**

While others have studied the (non-strategic) m-expert problem, as well as the strategic 1-expert problem, the authors are the first to study the strategic m-expert problem. This setting is well-motivated by applications such as forecasting competitions. While the algorithms presented by authors are not particularly novel, their application to the strategic m-expert problem requires non-trivial theoretical analysis, as well as some new ideas, such as (1) the sufficient condition for the perturbation distribution to guarantee approximate incentive compatibility in the modular setting, and (2) the adaptive regret bound for the WSU algorithm used as a subroutine for the submodular setting.

While I would normally consider the experimental results on real-world data as a strength, their impact is limited due to the lack of relevant comparisons (see Weaknesses for more details).

Finally the writing is clear, which makes the authors' contributions easy to understand.

**Weaknesses:**

The lack of any sort of lower bound is a weakness, especially since (as the authors point out) the loss function they consider is exp-concave, and thus the regret rates of Hedge in the non-strategic version of the problem where one expert is selected scale only logarithmically in the number of experts (and independently of the time horizon). While it may indeed be the case that going from constant-in-T to T^{1/2} regret rates is the price to pay for incentive compatibility in the m-expert problem, this submission presents no evidence either for or against this claim.

It would be nice to have provide some background on the current state of the (non-strategic) m-expert problem in the related work.

Finally, the experimental results would be strengthened considerably if (1) the theoretical rates and (2) the regret of the naive application of the WSU algorithm were also plotted. Currently, it is not easy to see how well the algorithms are actually doing without the relevant baselines. Additionally, it would be interesting to see how closely the empirical performance of the algorithms match their corresponding theoretical bounds.

**Questions:**

What are the runtimes of both algorithms?

What regret rates are obtained for the non-strategic m-experts problem with squared loss?

**Limitations:**

The authors have adequately addressed the limitations of their work.

---

> ### Author Rebuttal · Authors · 2023-08-09
>
> First, we would like to thank the reviewer for their constructive feedback and comments. We hope our responses below may help and perhaps convince the reviewer to raise their score. We are happy to answer any further questions during the author-reviewer discussion period.
>
> $\bullet$ **Q:** The lack of any sort of lower bound is a weakness, especially since (as the authors point out) the loss function they consider is exp-concave, and thus the regret rates of Hedge in the non-strategic version of the problem where one expert is selected scale only logarithmically in the number of experts (and independently of the time horizon). While it may indeed be the case that going from constant-in-$T$ to $T^{1/2}$ regret rates is the price to pay for incentive compatibility in the $m$-expert problem, this submission presents no evidence either for or against this claim.
>
> **A:** While there are lower bounds for the $m$-experts problem without the incentive compatibility property (and without using the fact that losses are quadratic) (see the response to your next question), neither the prior works nor our work provides lower bounds under the incentive compatibility assumption (and using the fact that losses are quadratic). That being said, we thought a lot about this and we conjecture that incurring an $\mathcal{O}(\sqrt{T})$ regret is inevitable in the strategic setting. In particular, approximately incentive-compatible algorithms typically incur an $\mathcal{O}(\sqrt{T})$ term in their regret bound only due to not being fully incentive-compatible. Therefore, even if the algorithm has better regret bounds in the non-strategic setting (i.e., the equivalent of Theorem 4 for FTPL) (e.g., $\mathcal{O}(\ln K)$ regret bound of Hedge as discussed in lines 361-363 of the paper), the overall regret would be $\mathcal{O}(\sqrt{T})$. We also came up with multiple fully incentive-compatible algorithms, however, it seems that obtaining regret bounds better than $\mathcal{O}(\sqrt{T})$ is at odds with the algorithm being fully incentive-compatible.
>
> $\bullet$ **Q:** It would be nice to have provide some background on the current state of the (non-strategic) $m$-expert problem in the related work.
>
> **A:** For the non-strategic setting with modular utilities, [10] proposed the Component Hedge (CH) algorithm and obtained a regret bound of $\sqrt{2m\ell^*\ln(\frac{K}{m})}+m\ln(\frac{K}{m})$ where $\ell^*$ is the cumulative loss of the best-chosen set in hindsight. They also gave a matching lower bound for this problem. Applying the same analysis as in Theorem 7 to the setting of the naive approach with $K \choose m$ meta-experts, we can show that the regret bound of WSU matches the aforementioned lower bound. [11] studied the FTPL algorithm with Gaussian noise distribution and provided an $\mathcal{O}(m\sqrt{T\ln (\frac{K}{m}}))$ regret bound for this setting. For the non-strategic setting with submodular utility functions, [13] proposed the online distorted greedy algorithm (with dual averaging algorithm as the algorithm $\mathcal{A}_i$ for $i=1,\ldots,m$) whose regret bound is $\mathcal{O}(\sqrt{mT\ln (\frac{K}{m})})$. More recently, [12] studied the $m$-experts problem under various different choices of the utility function (sum-reward, max-reward, pairwise-reward and monotone reward). In particular, for the setting with modular utilities (sum-reward), they proposed an algorithm that matches the optimal regret bound of the CH algorithm while being computationally more efficient. We will make sure to add this discussion to the "Related work" Section (Section 1.1) in the final version of the paper.
>
> $\bullet$ **Q:** The experimental results would be strengthened considerably if (1) the theoretical rates and (2) the regret of the naive application of the WSU algorithm were also plotted. Currently, it is not easy to see how well the algorithms are actually doing without the relevant baselines. Additionally, it would be interesting to see how closely the empirical performance of the algorithms match their corresponding theoretical bounds.
>
> **A:** We tried to implement the naive application of the WSU algorithm in our experiments. However, the number of meta-experts for our two experiments was $20 \choose 5$$=15504$ and $100\choose 5$$=75287520$ and the experiments did not finish (we waited for a few hours). In the "global" response to all the reviewers, we have included two new plots in which we have compared the running average of the regret of our proposed algorithms with that of the FiveThirtyEight aggregated predictions as the baseline. Please see the "global" response for more details.
>
> $\bullet$ **Q:** What are the runtimes of both algorithms?
>
> **A:** Both algorithms had similar runtime for our experiment which was around 3-4 minutes for each.
>
> $\bullet$ **Q:** What regret rates are obtained for the non-strategic $m$-experts problem with squared loss?
>
> **A:** While there exist prior works on the non-strategic $m$-experts problem (discussed above), none of these papers take the structure of the loss function (quadratic loss in our setting) into account. In order to obtain the $O(\ln K)$ regret bound for the $1$-expert problem (with squared loss) using the Hedge algorithm, the algorithm makes a single prediction $\sum_{i=1}^K \pi_{i,t}p_{i,t}$ at round $t\in[T]$ and its loss is $(\sum_{i=1}^K \pi_{i,t}p_{i,t}-r_t)^2$. In other words, choosing an expert $i\in[K]$ with probability $\pi_{i,t}$ at round $t$ is not good enough to obtain the improved $\mathcal{O}(\ln K)$ regret bound. Moving on to the $m$-experts problem, the main challenge for obtaining regret bounds better than $\mathcal{O}(\sqrt{T})$ is to decide how to aggregate the $K$ predictions as $m$ scalar values.

---

> > ### Comment · Reviewer_oxKD · 2023-08-11
> >
> > Thanks for your detailed reply. Please allow me to clarify my question about the runtime. In particular, I am asking about the theoretical runtime in Big O notation, as opposed to wall clock time.

---

> > > ### Author Response · Authors · 2023-08-11
> > > **Running Time of the Proposed Algorithms**
> > >
> > > Thanks for the clarification.
> > >
> > > At each round, the FTPL algorithm simply requires picking the $m$ experts (among the $K$ available experts) with the smallest noisy losses. We implemented this using a binary heap whose running time is $\mathcal{O}(K+m\ln K)$ per round. The online distorted greedy algorithm uses $m$ instances of the WSU algorithm for the $1$-expert problem and simply outputs the union of the experts chosen by these algorithms. At round $t\in[T]$, The WSU algorithm takes $\mathcal{O}(K)$ to compute the probabilities $\pi_{i,t}$ and pick one expert according to these probabilities. Therefore, the running time of the online distorted greedy algorithm is $\mathcal{O}(mK)$ per round.
> > >
> > > We are happy to answer any further questions.

---

> > > > ### Comment · Reviewer_oxKD · 2023-08-11
> > > >
> > > > Shouldn't the runtime of the online distorted greedy algorithm be random, due to the fact that it may be necessary to re-sample experts (as stated in Line 294)? If so, is $\mathcal{O}(mK)$ the *expected* per-round runtime?

---

> > > > > ### Author Response · Authors · 2023-08-11
> > > > >
> > > > > First, note that we did not require the algorithm to do resampling to obtain the theoretical bound. Therefore, the proven regret bound holds even if there are repeated experts that are chosen by the $m$ instances of the WSU algorithm. In this case, we can simply set the loss of the repeated experts to be 1. That being said, assume that at round $t\in[T]$, the WSU instance $\mathcal{A}\_j$ should pick an expert and the set of experts $\\{v\_{1,t},\dots,v\_{j-1,t}\\}$ are already chosen by $\mathcal{A}\_1,\dots,\mathcal{A}\_{j-1}$. In this case, the number of samples $\mathcal{A}_j$ needs to take to obtain a new expert follows a geometric distribution whose expected value is $\frac{1}{p}$ where $p$ is the probability of success (i.e., probability of picking one of the $K-j+1$ experts that are not already chosen). However, in practice, there are ways to simulate this step without actually performing repeated sampling. This question was addressed in Section 4.6.1 (title: "Avoiding duplicate actions") of [this paper](http://reports-archive.adm.cs.cmu.edu/anon/anon/home/ftp/usr0/ftp/2007/CMU-CS-07-171.pdf) who first introduced the online greedy algorithm for maximizing submodular functions.

---

> > > > > > ### Comment · Reviewer_oxKD · 2023-08-11
> > > > > >
> > > > > > Thanks for the reply. I encourage the authors to include such a discussion on runtime (as well as the related work on the non-strategic $m$-expert problem) in the revision. With that being said, most of my concerns have been addressed by the authors' replies, and so I will raise my score from 5 to 6.

---

> > > > > > > ### Author Response · Authors · 2023-08-11
> > > > > > >
> > > > > > > Thanks for taking the time to read our rebuttal, we are glad that our response has addressed your concerns. We will make sure to use the additional content page in the final version of the paper to include the discussion on runtime and also the related work on the non-strategic $m$-experts problem.

---

### Official Review · Reviewer_Aodd · 2023-07-05

**Soundness:** 3 good
**Presentation:** 4 excellent
**Contribution:** 3 good
**Rating:** 6
**Confidence:** 3

**Summary:**

This paper generalizes the problem of online binary prediction with expert advice, where at each round the learner can pick m>1 experts and the overall utility is a modular or submodular utility of the chosen experts. The experts are strategic and wish to be selected by the algorithm as often as possible hence they may misreport their beliefs about the events. They design algorithms that are incentive-compatible and achieve sublinear regret in the hindsight.
Previous work has studied this problem for m=1. In the case of m>1, previous work have focused on designing no-regret algorithms and did not take into consideration the incentive-compatibility issues.
Their algorithm builds on a prior work that studies the FTPL algorithm for the m-expert problem with modular utility functions and derive conditions for the perturbation function to guarantee approximate incentive compatibility. In particular, they show that FTPL while Gaussian perturbations is not incentive-compatible, Laplace or hyperbolic noise distribution guarantees approximate incentive-compatibility.

Moreover, they propose an algorithm that builds upon online monotone submodular maximization s.t. matroid constraints that takes m incentive-compatible for standard experts problem (i.e. m=1) and outputs their combined prediction. Compared to the first algorithm, this one achieves exact incentive-compatibility however, at the price of an extra \sqrt{m} term in the regret bound.


**Strengths:**

The paper is written nicely and is easy to follow. The problem seems to be well-motivated. I did not go through the proofs. The second algorithm is more straightforward, however, I think it is nice that they included it and compared it with their first algorithm.


**Weaknesses:**

It might be helpful to add a discussion on the regret lower bound for this problem.

**Questions:**

Minor comments:
The citations need to be fixed, they do not show the authors' name, need to use \citet.
Line 268, can you expand on why this holds?

Question : What is the lower bound for the regret of this problem? Can you give a lower bound that is stronger than the case for m=1?


**Limitations:**

Yes.

---

> ### Author Rebuttal · Authors · 2023-08-09
>
> First, we would like to thank the reviewer for their constructive feedback and comments. We hope our responses below may help and perhaps convince the reviewer to raise their score. We are happy to answer any further questions during the author-reviewer discussion period. Also, we have provided additional plots that better demonstrate the performance of our proposed algorithms. Please see the "global" response to all the reviewers for more details.
>
> $\bullet$ **Q:** It might be helpful to add a discussion on the regret lower bound for this problem. What is the lower bound for the regret of this problem? Can you give a lower bound that is stronger than the case for m=1?
>
> **A:** For the non-strategic setting with modular utilities (and without using the fact that losses are quadratic), [10] proposed the Component Hedge (CH) algorithm and obtained a regret bound of $\sqrt{2m\ell^*\ln(\frac{K}{m})}+m\ln(\frac{K}{m})$ where $\ell^*$ is the cumulative loss of the best-chosen set in hindsight. They also gave a matching lower bound for this problem. Applying the same analysis as in Theorem 7 to the setting of the naive approach with $K \choose m$ meta-experts, we can show that the regret bound of WSU matches the aforementioned lower bound. For the non-strategic setting with submodular utility functions (and without using the fact that losses are quadratic), [13] proposed the online distorted greedy algorithm (with dual averaging algorithm as the algorithm $\mathcal{A}_i$ for $i=1,\ldots,m$) whose $(1-\frac{c}{e})$-regret bound is $\mathcal{O}(\sqrt{mT\ln (\frac{K}{m})})$ and this is the best bound for submodular utilities. While there are lower bounds for the $m$-experts problem without the incentive compatibility property (and without using the fact that losses are quadratic) as discussed above, neither the prior works ([6,7]) nor our work provides lower bounds under the incentive compatibility assumption (and using the fact that losses are quadratic). That being said, we thought a lot about this and we conjecture that incurring an $\mathcal{O}(\sqrt{T})$ regret is inevitable in the strategic setting (and considering the fact that losses are quadratic). In particular, approximately incentive-compatible algorithms typically incur an $\mathcal{O}(\sqrt{T})$ term in their regret bound only due to not being fully incentive-compatible. Therefore, even if the algorithm has better regret bounds in the non-strategic setting (i.e., the equivalent of Theorem 4 for FTPL) (e.g., $\mathcal{O}(\ln K)$ regret bound of Hedge as discussed in lines 361-363 of the paper), the overall regret would be $\mathcal{O}(\sqrt{T})$. We also came up with multiple fully incentive-compatible algorithms, however, it seems that obtaining regret bounds better than $\mathcal{O}(\sqrt{T})$ is at odds with the algorithm being fully incentive-compatible.
>
> $\bullet$ **Q:** The citations need to be fixed, they do not show the authors' name, need to use citet.
>
> **A:** We will make sure to fix this in the final version of the paper.
>
> $\bullet$ **Q:** Line 268, can you expand on why this holds?
>
> **A:** We can write:
> \begin{equation*}
>     \mathbb{E}[\frac{1}{m}\sum_{t=1}^T\sum_{i\in S_t}\ell(b_{i,t},r_t)-\min_{S:|S|=m}\frac{1}{m}\sum_{t=1}^T\sum_{j\in S}\ell(b_{j,t},r_t)]=\mathbb{E}[\frac{1}{m}\sum_{t=1}^T\sum_{i\in S_t}\ell(p_{i,t},r_t)-\min_{S:|S|=m}\frac{1}{m}\sum_{t=1}^T\sum_{j\in S}\ell(p_{j,t},r_t)]+\mathbb{E}[\frac{1}{m}\sum_{t=1}^T\sum_{i\in S_t}\ell(b_{i,t},r_t)-\frac{1}{m}\sum_{t=1}^T\sum_{i\in S_t}\ell(p_{i,t},r_t)]+\mathbb{E}[\min_{S:|S|=m}\frac{1}{m}\sum_{t=1}^T\sum_{j\in S}\ell(p_{j,t},r_t)-\min_{S:|S|=m}\frac{1}{m}\sum_{t=1}^T\sum_{j\in S}\ell(b_{j,t},r_t)].
> \end{equation*}
> Let $S_1=\text{arg}\min_{S:|S|=m}\frac{1}{m}\sum_{t=1}^T\sum_{j\in S}\ell(b_{j,t},r_t)$ and $S_2=\text{arg}\min_{S:|S|=m}\frac{1}{m}\sum_{t=1}^T\sum_{j\in S}\ell(p_{j,t},r_t)$. We have:
> \begin{equation*}
>     \mathbb{E}[\frac{1}{m}\sum_{t=1}^T\sum_{i\in S_t}\ell(b_{i,t},r_t)-\min_{S:|S|=m}\frac{1}{m}\sum_{t=1}^T\sum_{j\in S}\ell(b_{j,t},r_t)]\leq\mathbb{E}[\frac{1}{m}\sum_{t=1}^T\sum_{i\in S_t}\ell(p_{i,t},r_t)-\min_{S:|S|=m}\frac{1}{m}\sum_{t=1}^T\sum_{j\in S}\ell(p_{j,t},r_t)]+\mathbb{E}[\frac{1}{m}\sum_{t=1}^T\sum_{i\in S_t}\ell(b_{i,t},r_t)-\frac{1}{m}\sum_{t=1}^T\sum_{i\in S_t}\ell(p_{i,t},r_t)]+\mathbb{E}[\frac{1}{m}\sum_{t=1}^T\sum_{j\in S_1}\ell(b_{j,t},r_t)-\frac{1}{m}\sum_{t=1}^T\sum_{j\in S_1}\ell(p_{j,t},r_t)]\leq \mathcal{O}(\sqrt{BT\ln (\frac{K}{m})})+\frac{1}{m}\sum_{t=1}^T\sum_{i\in S_t}\frac{4B}{\eta-2B}+\frac{1}{m}\sum_{t=1}^T\sum_{j\in S_1}\frac{4B}{\eta-2B}=\mathcal{O}(\sqrt{BT\ln (\frac{K}{m})})+\frac{8BT}{\eta-2B},
> \end{equation*}
> where the last inequality follows from line 265 and line 267 of the paper.

---

> > ### Comment · Reviewer_Aodd · 2023-08-16
> > **Responding to rebuttal**
> >
> > Thank you for your detailed response. I understand that addressing the lower bound is not a necessity for this paper and it can be studied in future work. It might be helpful to add your conjecture for the lower bound and raise it as an open question.

---

> > > ### Author Response · Authors · 2023-08-16
> > >
> > > Thanks for taking the time to read our response, we will make sure to use the additional content page in the final version of the paper to add the discussion about lower bounds.

---

### Official Review · Reviewer_xXJd · 2023-07-09

**Soundness:** 3 good
**Presentation:** 3 good
**Contribution:** 3 good
**Rating:** 6
**Confidence:** 3

**Summary:**

- The paper studies the problem of online prediction with expert advice. In their setting, a learner at each round selects $m$ experts, and the loss is determined by a submodular or modular function of the beliefs reported by the selected experts. Each expert, being strategic, may intentionally misrepresent their beliefs to maximize their chances of being selected in the next round. The goal of the learner is to design no-regret algorithms that incentive experts to report their beliefs truthfully (also called IC).

- To tackle this problem, the authors first present an inefficient reduction to the case when $m=1$. Subsequently, they propose two efficient algorithms that are for the specific types of loss functions considered.

    - In the case of the modular loss function, the authors investigate Follow-The-Perturbed-Leader (FTPL) and propose a general condition for the perturbation distribution that ensures approximate IC.

    - For the submodular loss function,they propose a simpler algorithm named the online distorted greedy algorithm. This algorithm not only guarantees exact IC but also attains the optimal approximation ratio.

**Strengths:**

- The paper studies an important and generic problem of online prediction with strategic experts that may find its applications in many scenarios.
- The proposed algorithms in this paper are not only easy to implement and also enjoy good theoretical guarantees of being (approximate) IC and no regret.

**Weaknesses:**

1. Several claims may have some issues, see eg Questions 3, 5, 6.
2. The presentation could be improved. For instance, Section 1.2 titled "Contributions" seems to contain some elements of background and motivations which might be more appropriately placed in the introduction of the paper. Moreover, it would be beneficial to include more discussions following important definitions (like Def 1) and theorems to provide more intuitions.
3. In the experimental section, the authors plot the regret curve that is not normalized. This makes it hard to interpret, as unnormalized regret curves don't explicitly demonstrate the rate at which the algorithms learn. Additionally, to reflect the difference between reported and true beliefs, a uniformly random value within the range guaranteed by Theorem 3 is added, but this theorem doesn't ensure a uniform distribution for this difference. Finally, incorporating a comparison of the proposed algorithms with and aggregated predictions of FiveThirtyEight could potentially enrich the analysis.

**Questions:**

1. Regarding Def 1:
    - Is this a new concept being proposed, or has it previously been studied in previous works?
    - The experts are long-standing in this setting, but the focus of the paper is on myopic IC. Although the authors mentioned that the analysis of FTPL could be extended to the more general setting of maximizing a conic combination of probabilities of being chosen at all subsequent rounds, is it also true for the online distorted greedy algorithm? What's the main modification/challenge of extending to non-myopic experts?
    - Would achieving IC become impossible if the definition is for all $r_t$, rather than the expectation over Bern( $b_{i,t}$ )?

2. In the "forecasting competitions" example described in Section 3.1, it seems that the loss function is not captured by the modular function studied in this paper, given it is normalized by $|S_t|$ rather than $m$.

3. About Thm 1, could you clarify why the reduction preserves IC? In particular:
    - How is the belief of each meta-expert defined in this context? Is the new loss function $\ell_S$ still proper? How to define IC when the belief of each expert is multi-dimensional?
    - Even though WSU is IC for the $m=1$ problem, it only guarantees every single expert reports truthfully when fixing other's reports. In particular, it doesn't rule out the possibility that a group of experts form a coalition and misreport together to increase the sum of their probabilities of being selected, which is exactly what would happen when a single expert misreports in the $m$-expert problem -- it results in the simultaneous misreporting of $\binom{K}{m-1}$ meta-experts.

4. Thm 3 guarantees approximate IC in the sense of $|p^\star\_{i,t}-b\_{i,t}|$ being small. However, a more common notion of approximate IC is through the difference of utility, which, applies to this setting, is the difference in $\pi_{i,t+1}$ when $p^\star_{i,t}$ and $b_{i,t}$ are reported respectively. Could the authors include a discussion about this alternative notion?

5. Combining Thm 3 and Thm 5, it seems that as $B$ approaches 0, the FTPL algorithm becomes more IC and also yields a smaller regret. However, this contradicts the intuition that if the learner's decisions are entirely noise-driven (i.e., $B=0$), the regret should be high. Could the authors comment on this?

6. On lines 297-298, you mention that the $(1-\frac{c_f}{e})$ approximation factor is optimal. Is this claim supported by a lower bound? If so, could you point to the relevant references?

**Limitations:**

Yes.

---

> ### Author Rebuttal · Authors · 2023-08-09
>
> First, we would like to thank the reviewer for their constructive feedback and comments. We hope our responses below may help and perhaps convince the reviewer to raise their score. We are happy to answer any further questions during the author-reviewer discussion period. Also, given the size limit for the rebuttal, we have included the responses to the rest of your questions at the end of the "global" response to all the reviewers.
>
> $\bullet$ **Q:** The presentation could be improved. For instance, Section 1.2 titled "Contributions" seems to contain some elements of background and motivations which might be more appropriately placed in the introduction of the paper. Moreover, it would be beneficial to include more discussions following important definitions (like Def 1) and theorems to provide more intuitions.
>
> **A:** Thanks for your suggestions. We will make sure to use the additional page in the final version of the paper and provide intuitions for the definition of incentive compatibility (Definition 1) and add more discussions following our theorems. Also, we will revise the "Contributions" section (Section 1.2) and move the motivations to the "Introduction" section (Section 1) and the literature review to the "Related work" section (Section 1.2).
>
> $\bullet$ **Q:** In the experimental section, the authors plot the regret curve that is not normalized. This makes it hard to interpret, as unnormalized regret curves don't explicitly demonstrate the rate at which the algorithms learn. Additionally, to reflect the difference between reported and true beliefs, a uniformly random value within the range guaranteed by Theorem 3 is added, but this theorem doesn't ensure a uniform distribution for this difference. Finally, incorporating a comparison of the proposed algorithms with and aggregated predictions of FiveThirtyEight could potentially enrich the analysis.
>
> **A:** In the plots attached to the "global'' response to all reviewers, we have plotted the running average of regret $\frac{1}{t}\mathbb{E} \big[\max_{S\subseteq [K]:|S|=m}\sum_{\tau=1}^t f_{\tau}(S)-\sum_{\tau=1}^t f_{\tau}(S_{\tau})\big]$ for $1\leq t\leq T$. In other words, we have normalized the regret by the horizon length (please let us know if we misunderstood your point about normalized regret curves and you had a different type of normalization in mind). Also, we have included the plot for running average regret of the aggregated prediction of FiveThirtyEight to highlight the superior performance of our proposed algorithms. Please see the "global" response to all the reviewers for more details. As we have shown in the proof of Theorem 3, at round $t\in [T]$, expert $i\in [K]$ needs access to the reports of other experts $p_{j,t}$ for $j\neq i$ to be able to compute the optimal reported belief $p_{i,t}^*$. Given that this is generally not possible, we have added a uniformly random value in the range derived in Theorem 3 to model the difference between the reported and true beliefs. Note that this uniformly random value adversely affects the performance of the FTPL algorithm because the algorithm receives the noisy reported beliefs, however, its performance is evaluated based on the true beliefs.
>
> $\bullet$ **Q:** Regarding Def 1: a) Is this a new concept being proposed, or has it previously been studied in previous works? b) The experts are long-standing in this setting, but the focus of the paper is on myopic IC. Although the authors mentioned that the analysis of FTPL could be extended to the more general setting of maximizing a conic combination of probabilities of being chosen at all subsequent rounds, is it also true for the online distorted greedy algorithm? What's the main modification/challenge of extending to non-myopic experts? c) Would achieving IC become impossible if the definition is for all $r_t$, rather than the expectation over $\text{Bern}(b_{i,t})$?
>
> **A:** Definition 1 was first introduced by [6]. Given that the WSU algorithm of [6] is only incentive-compatible in the myopic setting, we focused on this definition of incentive compatibility to be able to use the WSU algorithm as a sub-routine in the online distorted greedy algorithm. However, as we mentioned in the paper, it is easy to show that FTPL is also approximately incentive compatible with respect to the non-myopic incentive structure (we just need to repeat the same analysis as the one in the proof of Theorem 3 multiple times for the probability of being chosen in each of the subsequent rounds). If we use FTPL as the algorithm $\mathcal{A}_i$ for $i=1,\ldots,m$ in the online distorted greedy algorithm, the online distorted greedy algorithm would be approximately incentive-compatible in the non-myopic incentive structure as well. We are not sure if we understand your last question (part c), it would be great if you could clarify it a bit further.
>
> $\bullet$ **Q:** In the "forecasting competitions" example described in Section 3.1, it seems that the loss function is not captured by the modular function studied in this paper, given it is normalized by $|S_t|$ rather than $m$.
>
> **A:** We defined the modular utility function as $f_t(S_t)=\frac{1}{m}\sum_{i\in S_t}(1-\ell_{i,t})=\frac{|S_t|}{m}-\frac{1}{m}\sum_{i\in S_t}\ell_{i,t}$ to make sure that the utility function is monotone and modular so that the online distorted greedy algorithm is applicable. If the cardinality of set $S_t$ is equal to $m$ (which is true for $S_t$ output by our proposed algorithms), the utility $f_t(S_t)$ is simply one minus the average loss of chosen experts. This is the utility function considered for the "forecasting competitions" example in Section 3.1.

---

> > ### Comment · Reviewer_xXJd · 2023-08-15
> >
> > Thank you very much for the detailed response. Given that most of my concerns have been addressed, I'm happy to raise my score from 5 to 6.

---

> > > ### Author Response · Authors · 2023-08-15
> > >
> > > Thanks for taking the time to go over our response, we are glad that we were able to address most of your concerns.

---

### Official Review · Reviewer_9kMc · 2023-07-09

**Soundness:** 4 excellent
**Presentation:** 3 good
**Contribution:** 3 good
**Rating:** 6
**Confidence:** 3

**Summary:**

The paper broadly focuses on the design and analysis of no-regret and incentive-compatible algorithms for the m-experts problem (pick a subset $S_t \subseteq [K]$ of $m$ experts in each round $t$ to obtain a utility value $f_t(S_t)$ which is a function of the losses $\ell_{i,t} \in [0,1]$ for all $i \in S_t$), with specific utility functions (on the space of m-sized subsets of experts) in each round which are either modular or sub-modular.
* They try to improve on the inefficient baseline of using the no-regret, incentive-compatible WSU algorithm (Freeman et all 2020) on the set of $\binom{K}{m}$ meta-experts corresponding to sets of $m$ experts.
* They analyze the FTPL algorithm in the modular utility function case where the perturbation distribution is zero-mean symmetric from an exponential family. They propose a sufficient-condition for this distribution (related to bounded hazard rate) which implies that FTPL would be no-regret and approximately incentive-compatible. In particular, they show that this condition is satisfied for the Laplace distribution and the symmetric hyperbolic distribution etc.
* They propose and analyze an "online distorted greedy algorithm" (based on earlier work by (Harvey et al. 2020) on online submodular maximizination) for the submodular utility function case. They show that the algorithm has sublinear $\alpha$-regret bounds and is $\alpha$-approximately-incentive compatible. where $\alpha$ may be $ < 1$ for general submodular functions (depending on the "curvature") while $\alpha = 1$ in the modular case. This of course implies that the algorithm has no-regret in the usual sense and is fully incentive compatible in the modular case, but at the expense of more computation and a slightly worse regret bound compared to FTPL.




**Strengths:**

* The online $m$-experts problem is well-motivated in the introduction, as is the importance of incentive compatibility for the problem.
* The algorithms proposed (FTPL and online distorted greedy) are natural for the problem and build nicely on previous work. All the appropriate details are given, along with the code and experiments.
* The proposed sufficiency condition for the perturbation distribution of FTPL is fairly general, and may be of independent interest in the study of incentive compatibility.
* Sufficient proof-sketches are given for quite a few of the results in the main paper itself, and more detail is provided in the supplementary material.



**Weaknesses:**

* The FTPL analysis applies to a single modular utility function defined in the paper (Line 136). Of course, this utility function is fairly natural for the problem.
* The necessity of "Condition 1" for getting approximate-incentive-compatibility with FTPL is not explored.
* The experimental evaluation provided is fairly limited (of course, the thrust of the work is theoretical).


**Questions:**

None

**Limitations:**

Not applicable.

---

> ### Author Rebuttal · Authors · 2023-08-09
>
> First, we would like to thank the reviewer for their constructive feedback and comments. We hope our responses below may help and perhaps convince the reviewer to raise their score. We are happy to answer any further questions during the author-reviewer discussion period.
>
> $\bullet$ **Q:** The FTPL analysis applies to a single modular utility function defined in the paper (Line 136). Of course, this utility function is fairly natural for the problem.
>
> **A:** While the FTPL algorithm is only applicable for the setting with modular utility functions, we can use FTPL as the algorithm $\mathcal{A}_i$ for $i=1,\ldots,m$ in the online distorted greedy algorithm, and obtain results for the submodular utility setting as well. In this case, the online distorted greedy algorithm would be only approximately incentive-compatible.
>
> $\bullet$ **Q:** The necessity of "Condition 1" for getting approximate-incentive-compatibility with FTPL is not explored.
>
> **A:** We believe that Condition 1 is not necessary to achieve approximate incentive compatibility. For example, it is well-known that the FTPL algorithm with Gumbel noise distribution is equivalent to the Hedge algorithm. [7] showed that the Hedge algorithm is approximately incentive-compatible for the $1$-expert problem. However, as we have mentioned in lines 248-249 of the paper, Condition 1 does not hold for the Gumbel distribution.
>
> $\bullet$ **Q:** The experimental evaluation provided is fairly limited (of course, the thrust of the work is theoretical).
>
> **A:** We have provided additional plots that better demonstrate the performance of our proposed algorithms. Please see the "global" response to all the reviewers for more details.

---

> > ### Comment · Reviewer_9kMc · 2023-08-21
> > **Response**
> >
> > Thank you for the clarifications in this and in the  global rebuttal. I do believe incorporating some of those points into the final version of the paper will certainly make it better. I will keep the score for now but will take the clarifications in consideration if further discussion with the AC becomes necessary.

---

> > > ### Author Response · Authors · 2023-08-21
> > >
> > > Thanks for taking the time to read our response, we will make sure to use the additional content page in the final version of the paper and incorporate the points you mentioned in your review (particularly the necessity of Condition 1).

---

### Author Rebuttal · Authors · 2023-08-09

In the attached file, we have included two additional figures plotting the running average of regret $\frac{1}{t}\mathbb{E} \big[\max_{S\subseteq [K]:|S|=m}\sum_{\tau=1}^t f_{\tau}(S)-\sum_{\tau=1}^t f_{\tau}(S_{\tau})\big]$ for $1\leq t\leq T$ for our proposed algorithms and also the FiveThirtyEight aggregated predictions in the $K=20$ and $K=100$ settings. Note that while our proposed algorithms choose $m$ predictions at each round $t\in[T]$, the FiveThirtyEight aggregated prediction $\bar{p}_t\in [0,1]$ is a single scalar value and we measure its loss as $(\bar{p}_t-r_t)^2$. As could be seen in the plots, while the regret of all three algorithms converge to zero as $t$ gets larger, both our proposed algorithms have a superior performance compared to that of the FiveThirtyEight prediction. It is also noteworthy that the overall regret of both our algorithms is way better than the corresponding theoretical bounds proved in the paper. Given the huge distance between the theoretical bounds and the regret curves, we decided not to include them.

# Rest of Response to Reviewer xXJd:

$\bullet$ **Q:** About Thm 1, could you clarify why the reduction preserves IC?

**A:** Note that for the naive approach discussed in Section 3.2, we still define incentive compatibility with respect to individual experts (instead of the $K \choose m$ meta-experts). To be precise, we define $\pi_{i,t}=\sum_{S:|S|=m,i\in S}\pi_{S,t}$. For both choices of the loss function ($\ell_S=\frac{1}{m}\sum_{j\in S}\ell_{j,t}$ and $\ell_S=\prod_{j\in S}\ell_{j,t}$ where $i\in S$), the loss of the set is linear in $\ell_{i,t}$ and given the fact that the loss function is proper, we can show that in such cases, the WSU algorithm applied to $K \choose m$ meta-experts is incentive-compatible. For example, for the setting with modular utility function ($\ell_S=\frac{1}{m}\sum_{j\in S}\ell_{j,t}$), we can show the following:
\begin{equation*}
\pi_{i,t+1}=\sum_{S:|S|=m,i\in S}\pi_{S,t+1}=\pi_{i,t}(1-\frac{\eta}{m} L_{i,t})-\frac{\eta}{m} \sum_{s\neq i}(\sum_{S:|S|=m,\{i,s\}\subseteq S}\pi_{S,t})\ell_{s,t}.
\end{equation*}
Given that $\pi_{i,t+1}$ is linear in $L_{i,t}$, $L_{i,t}=\ell_{i,t}-\sum_{j=1}^K \pi_{j,t}\ell_{j,t}$ is linear in $\ell_{i,t}$, and the loss function is proper, we can conclude that incentive compatibility holds in this setting. Thanks for your question, we will make sure to clarify this point in the final version of the paper.

$\bullet$ **Q:** Thm 3 guarantees approximate IC in the sense of $|p_{i,t}^*-b_{i,t}|$ being small. However, a more common notion of approximate IC is through the difference of utility, which, applies to this setting, is the difference in $\pi_{i,t+1}$ when $p_{i,t}^*$ and $b_{i,t}$ are reported respectively. Could the authors include a discussion about this alternative notion?

**A:** As we have shown in the proof of Theorem 3, the expected utility of expert $i$ (according to her belief $b_{i,t}$) at round $t$ (i.e., her probability of being chosen at round $t+1$) is $(1-b_{i,t})F_0(-p_{i,t}^2)+b_{i,t}F_1(-(1-p_{i,t})^2)$, where $Y_0=\eta \gamma_{i,t} + L - X_0^{(t)}$, $Y_1=\eta \gamma_{i,t} + L - X_1^{(t)}$, $f_{0}(Y_0)\propto \text{exp}\Big(-\nu\big(\frac{Y_0-(L-X_0^{(t)})}{\eta}\big)\Big)$, and $f_{1}(Y_1)\propto \text{exp}\Big(-\nu\big(\frac{Y_1-(L-X_1^{(t)})}{\eta}\big)\Big)$. Also, $F_0$ and $F_1$ denote the cdf corresponding to $f_0$ and $f_1$ respectively. Given that $F_0$ and $F_1$ are Lipschitz (because $f_0$ and $f_1$ are bounded), the difference in $\pi_{i,t+1}$ when $p_{i,t}^*$ and $b_{i,t}$ are reported can be bounded as a constant times $|p_{i,t}^*-b_{i,t}|$ and therefore, this alternative notion of incentive-compatibility is satisfied as well.

$\bullet$ **Q:** Combining Thm 3 and Thm 5, it seems that as $B$ approaches 0, the FTPL algorithm becomes more IC and also yields a smaller regret. However, this contradicts the intuition that if the learner's decisions are entirely noise-driven (i.e., $B=0$), the regret should be high. Could the authors comment on this?

**A:** In this paper, we focused on zero-mean symmetric noise distributions from the exponential family, i.e., $f(\gamma)\propto \text{exp}(-\nu(\gamma))$. Given that $|\nu'(z)|\leq B$, if $B=0$, $\nu(\cdot)$ needs to be constant which corresponds to the uniform distribution (note that in this case, the decisions are not entirely noise-driven). However, uniform distribution does not belong to the exponential family. Therefore, $B>0$. As we have mentioned in the proof of Theorem 3, we have:
\begin{equation*}
    p_{i,t}^*=\frac{b_{i,t}}{b_{i,t}+(1-b_{i,t})A},
\end{equation*}
where $A=\text{exp}\big(\nu(\frac{-(1-p_{i,t})^2-(L-X_1^{(t)})}{\eta})-\nu(\frac{-p_{i,t}^2-(L-X_0^{(t)})}{\eta})\big)$. Ideally, we want $A$ to be as close to 1 as possible because if $A=1$, $p_{i,t}^*=b_{i,t}$ and the algorithm would be incentive-compatible. In order to ensure approximate incentive compatibility, the pdf of the noise distribution $f$ needs to be such that $\frac{f(z)}{f(z+1)}$ does not grow to infinity for very large $z$. One way to enforce this condition is via a Lipschitzness assumption on $\ln f$. Condition 1 implies that $\ln f$ is $B$-Lipschitz. That is why smaller values of $B$ lead to better approximate incentive compatibility which in turn results in smaller regret bounds (given that the term $4TC$ appears in the regret bound where $C$ is the bound on the approximate incentive compatibility derived in Theorem 3).

$\bullet$ **Q:** On lines 297-298, you mention that the $(1-\frac{c_f}{e})$ approximation factor is optimal. Is this claim supported by a lower bound? If so, could you point to the relevant references?

**A:** You can find the reference paper below, we will make sure to include this in lines 297-298 of the paper:

Sviridenko, M., Vondrák, J., \& Ward, J. (2017). Optimal approximation for submodular and supermodular optimization with bounded curvature. Mathematics of Operations Research, 42(4), 1197-1218.

---

### Decision · Program_Chairs · 2023-09-21

**Decision:**

Accept (poster)

**Comment:**

The paper discusses expert advice settings, under the assumption that experts (agents) are strategic.
All reviewers made a terrific job by delving deep into the paper, and have raised several point which were discussed with the authors. Eventually, all reviewers are satisfied with the responses of the authors and with the paper's contribution, and unanimously vote for acceptance.